# The GRPO Tax is Smaller Than You Think: A Longitudinal Study of Capability Preservation During Reasoning Training

## Abstract

Reinforcement learning methods such as Group Relative Policy Optimization (GRPO) have emerged as a dominant paradigm for training language models to reason, following the success of DeepSeek-R1. A persistent concern is the "alignment tax": the degradation of general-purpose capabilities that accompanies post-training optimization for a specific objective. We conduct the first dense-checkpoint longitudinal study of capability evolution during GRPO training across five instruction-tuned models from four families (Qwen, Phi, Gemma, and Llama) at the 1.5B to 3.8B parameter scale, with a supplemental 7B QLoRA extension. Each model is evaluated at up to 75 training checkpoints across 13 benchmarks. Under single-epoch LoRA-based training, we establish that the GRPO tax is substantially smaller than commonly assumed: 85% of non-target capabilities remain within ±2% of their baseline values, 8% improve, and only 7% degrade. A series of supplemental analyses constrain and validate this finding. Three-seed verification on Qwen2.5-1.5B-Instruct confirms low cross-run variance with mean standard deviation of 0.004 across benchmarks. The 7B QLoRA extension preserves all 12 non-target capabilities with mean absolute delta of 0.003. A full-parameter fine-tuning ablation reduces preservation to 8/12 on Qwen2.5-1.5B-Instruct without catastrophic collapse, showing the preservation advantage persists beyond the LoRA constraint, while a controlled ablation that disables GRPO's group-relative reward normalization leaves preservation unchanged at 10/12, localizing the effect to GRPO's on-policy optimization rather than to the normalization itself. A matched-dataset DPO comparison on GSM8K-derived preference pairs confirms that DPO acts as a more conservative optimizer than GRPO independently of the training distribution, with standard deviation of $\Delta_k$ at 0.007 for DPO versus 0.017 for GRPO at 1.5B. An expanded $N$=150 safety audit demonstrates that the previously hypothesized cross-family safety divergence does not reach statistical significance, with Gemma and Llama exhibiting $-1.3\%$ and $-2.0\%$ safety shifts whose 95% confidence intervals overlap zero. A five-epoch continuation on two model families shows single-epoch training to be a uniformly safe stopping point, while the degree of degradation under extended training is model-dependent: Qwen2.5-1.5B preservation decays to 4/12 by epoch 5 whereas Llama-3.2-3B retains 9/12. We further show, across all five models, that the preservation property is not an artifact of a weak optimization setup or a saturated task: a larger group size ($G = 8$) yields stronger target gains of 12.5 to 16.5 points while preserving 9 to 11 of 12 capabilities, and GRPO on the harder MATH task improves the target by 5.6 to 9.4 points while preserving 11 to 13 of 13 benchmarks. Under flexible answer extraction the target improves by 8.5 to 12.5 points for every family at $G = 4$, confirming that the modest strict-extraction gains are a format-mismatch artifact rather than a learning failure. At matched target accuracy, on-policy GRPO incurs lower non-target KL than off-policy DPO and SFT, consistent with the on-policy KL-minimality account of Shenfeld et al. (2025). Within this regime, single-epoch LoRA or QLoRA GRPO with verifiable rewards on mathematical reasoning for instruction-tuned models at the 1.5B to 7B scale, these results challenge the prevailing narrative of catastrophic capability loss; we delimit the conclusions to this regime rather than to the GRPO tax in general.

# 1 Introduction

The release of DeepSeek-R1 (DeepSeek-AI, 2025) demonstrated that reinforcement learning, specifically Group Relative Policy Optimization (GRPO), can elicit sophisticated reasoning capabilities in large language models without explicit supervision of reasoning chains. This result catalyzed widespread adoption of RL-based post-training pipelines for reasoning, with subsequent work applying GRPO and related methods to domains ranging from code generation to scientific problem-solving (Yu et al., 2025).

A central concern is the alignment tax: optimizing a model for one objective tends to degrade performance on others (Lin et al., 2024a). This phenomenon has been documented for RLHF (Ouyang et al., 2022), safety alignment (Huang et al., 2025), and supervised fine-tuning (Li et al., 2024), but the specific dynamics of capability change during GRPO training remain underexplored.

Existing studies use a pre-post evaluation paradigm that obscures the temporal dynamics of capability change: when does degradation begin, how fast does it progress, and is there an optimal early-stopping point? Furthermore, most analyses focus on a single model family, making it difficult to distinguish model-specific artifacts from generalizable phenomena.

In this work, we address these gaps through a dense-checkpoint longitudinal study. We train five instruction-tuned models from four families (Qwen2.5-1.5B-Instruct, Qwen2.5-3B-Instruct, Phi-3.5-mini-instruct, Gemma-2-2B-it, and Llama-3.2-3B-Instruct) using GRPO on the GSM8K mathematical reasoning dataset (Cobbe et al., 2021), saving model checkpoints every 50 training steps. Each of the resulting 380 checkpoints (plus 5 baselines) is evaluated on 13 benchmarks spanning mathematical reasoning (Cobbe et al., 2021), general knowledge (Hendrycks et al., 2021), commonsense reasoning (Zellers et al., 2019), scientific reasoning (Clark et al., 2018), truthfulness (Lin et al., 2022), coreference resolution (Sakaguchi et al., 2020), instruction following, summarization, translation, code generation, safety, creative writing, and conversational quality. This produces over 5,616 individual evaluation data points, enabling fine-grained analysis of capability trajectories during training.

Our contributions are sixfold. First, within the single-epoch LoRA-based training regime at the 1.5B to 7B parameter scale, we establish that the GRPO tax on non-reasoning capabilities is substantially smaller than the prevailing narrative suggests: across 60 model-benchmark pairs at the 1.5B–3.8B scale (excluding the target task), 85% of capabilities remain within ±2% of baseline, 8% improve, and only 7% degrade beyond the 2% threshold. Three-seed verification on Qwen2.5-1.5B-Instruct yields mean cross-seed standard deviation of 0.004, and a supplemental 7B QLoRA extension preserves all 12 non-target capabilities with mean absolute delta of 0.003. Second, controlled mechanism ablations indicate that the preservation advantage is a property of GRPO's on-policy optimization rather than of its group-relative reward normalization or the LoRA constraint: on identical model and data, GRPO preserves 10/12 non-target capabilities while a supervised fine-tuning baseline preserves 8/12, a full-parameter GRPO ablation preserves 8/12 without catastrophic collapse, and disabling the group-relative normalization leaves preservation unchanged at 10/12. Third, a matched-dataset comparison of GRPO and DPO on GSM8K-derived preference pairs confirms that DPO acts as a more conservative optimizer with smaller capability variance than GRPO (standard deviation of $\Delta_k$ is 0.007 versus 0.017 at 1.5B; 0.005 versus 0.009 at 3B), eliminating the dataset confound from our main comparison. Fourth, an expanded safety audit at $N=150$ prompts sampled from HarmBench and AdvBench demonstrates that previously hypothesized cross-family safety divergences do not reach statistical significance, with Gemma and Llama 95% binomial confidence intervals overlapping zero. Fifth, a five-epoch continuation on two families (Qwen2.5-1.5B and Llama-3.2-3B) shows that single-epoch training is a uniformly safe stopping point, while the degree of degradation under extended training is model-dependent: Qwen2.5-1.5B preservation decays from 10/12 at epoch 1 to 4/12 by epoch 5, whereas Llama-3.2-3B retains 9/12 through five epochs. Sixth, we show that the preservation property is robust to the strength of the optimization setup and the choice of target task: across all five models, a larger group size ($G = 8$) yields stronger target gains while preserving 9 to 11 of 12 capabilities, and GRPO on the harder MATH task improves the target by 5.6 to 9.4 points while preserving 11 to 13 of 13 benchmarks; we further connect the finding to the on-policy account of Shenfeld et al. (2025) by formalizing a per-benchmark capability-drift bound, validating it empirically, and showing that at matched target accuracy on-policy GRPO incurs lower non-target KL divergence than off-policy DPO and SFT.

**Scope of our claims.** We delimit these conclusions deliberately. Our findings concern single-epoch, LoRA- and QLoRA-based GRPO with verifiable binary rewards on mathematical-reasoning target tasks (GSM8K and MATH), applied to instruction-tuned models in the 1.5B to 7B parameter range. We do not claim that the GRPO tax is small in general. Full-parameter training at larger scales, non-mathematical or multi-task objectives, training horizons beyond a single epoch, and models larger than 7B may exhibit different dynamics; where our supplemental experiments probe these axes (full fine-tuning at 1.5B, the five-epoch continuation, and the 7B QLoRA extension) we report the regime boundaries explicitly rather than extrapolating past them. The contribution is a precise, multi-method characterization of capability preservation within this regime, not a universal statement about reinforcement learning for reasoning.

## 2 Related Work

**Reinforcement Learning for Reasoning.** The application of reinforcement learning to enhance reasoning in language models has accelerated following the release of DeepSeek-R1 (DeepSeek-AI, 2025), which demonstrated that GRPO with verifiable rewards can elicit chain-of-thought reasoning without supervised reasoning traces. Subsequent work has extended GRPO to diverse domains including vulnerability detection (Simoni et al., 2025), medical reasoning (Pan et al., 2025), and multi-turn planning (Hu et al., 2025). Algorithmic improvements to GRPO have been proposed through scaffolded training curricula (Zhang et al., 2025), entropy-weighted reward shaping (Tan & Pan, 2025), and soft-token policy optimization (Zheng & Lee, 2025). The DAPO framework (Yu et al., 2025) demonstrated GRPO at scale, achieving competitive performance on AIME 2024 benchmarks.

**The Alignment Tax.** The observation that post-training optimization degrades non-target capabilities was first systematically studied by Lin et al. (2024a), who coined the term "alignment tax" and demonstrated monotonic capability degradation across NLP benchmarks during RLHF training. Huang et al. (2025) extended this analysis to reasoning models, identifying a trade-off between reasoning capability and safety alignment in the sequential production pipeline for large reasoning models. Lu et al. (2024b) proposed online merging optimizers to mitigate the alignment tax by integrating RL policy and SFT models at each optimization step. More recently, Sun et al. (2025) introduced orthogonal gradient projection to constrain safety updates to the null space of general capability gradients.

**Catastrophic Forgetting in Post-Training.** Catastrophic forgetting during LLM fine-tuning has been studied from multiple perspectives. Li et al. (2024) demonstrated that sharpness-aware minimization can flatten the loss landscape to reduce forgetting during supervised fine-tuning. Wolczyk et al. (2024) conceptualized the interplay between pre-trained capabilities and RL-specific adaptations as a forgetting problem, showing that standard retention techniques can preserve pre-trained knowledge. Lai et al. (2025) provided a comparative analysis showing that reinforcement fine-tuning naturally mitigates forgetting relative to supervised fine-tuning in continual post-training settings. The CapTrack framework (Thede et al., 2026) proposed a capability-centric taxonomy for evaluating forgetting across behavioral dimensions, though their analysis was limited to supervised and DPO post-training without GRPO coverage.

**Capability Evaluation During Training.** Several recent works have analyzed how capabilities evolve during RL training. Yao et al. (2025) identified a two-stage dynamic in RLVR training, with early exploitation followed by exploration. Wang et al. (2026) showed that model weights and output probabilities evolve linearly with RLVR training steps. Yue et al. (2025) controversially argued that RLVR does not create fundamentally new reasoning patterns but rather improves sampling efficiency over the base model's existing distribution. However, these studies either focus on the target reasoning task or adopt a coarse-grained evaluation protocol. Our work complements this literature by providing the first dense-checkpoint, multi-benchmark, cross-family longitudinal analysis specifically focused on non-target capability preservation during GRPO training.

Most directly related to our findings, Shenfeld et al. (2025) propose the "RL's Razor" hypothesis: the degree of forgetting during fine-tuning is governed by the KL divergence between the fine-tuned and base policy measured on the new task, and on-policy reinforcement learning is implicitly biased toward the KL-minimal solution among those that solve the task, which explains why it forgets less than supervised fine-tuning. Our

study was conducted independently and reaches the same conclusion that KL divergence governs forgetting, through a dense-checkpoint multi-method analysis rather than a controlled forgetting benchmark. We extend their account in three respects: we operationalize the KL-forgetting link as a per-benchmark capability-drift bound (Equation (2)); we validate that bound empirically across thirteen benchmarks (Section 5.14); and we report a direct test of the KL-minimality prediction across GRPO, DPO, and SFT at matched target accuracy (Section 5.17). We present our GRPO-specific gradient-gating property (Lemma 1) as complementary to their on-policy account rather than as a competing explanation.

**Direct Preference Optimization.** DPO (Rafailov et al., 2023) has emerged as a popular alternative to RL-based alignment, offering simpler training dynamics by directly optimizing the policy on preference pairs without a separate reward model. Comparative analyses of DPO and RL methods have examined aspects such as generalization (Lin et al., 2024b), length bias (Lu et al., 2024a), and robustness (Wu et al., 2024). Our work contributes to this comparative literature by examining how DPO and GRPO differ in their impact on non-target capabilities during training.

## 3 Methodology

### 3.1 Problem Formulation

Let $\pi_\theta$ denote a language model parameterized by $\theta$, initialized from an instruction-tuned checkpoint $\pi_{\theta_0}$. GRPO training optimizes $\theta$ to maximize a reward function $r(\mathbf{y} \mid \mathbf{x})$ on a task-specific dataset $\mathcal{D} = \{(\mathbf{x}_i, \mathbf{y}_i^*)\}_{i=1}^N$, where $\mathbf{x}_i$ is a prompt and $\mathbf{y}_i^*$ is the verifiable ground truth. We define the capability profile of $\pi_\theta$ as its performance vector across $K$ evaluation benchmarks:

$$\mathbf{c}(\theta) = [s_1(\pi_\theta), s_2(\pi_\theta), \ldots, s_K(\pi_\theta)] \in [0,1]^K$$

where $s_k(\pi_\theta)$ denotes the score of $\pi_\theta$ on benchmark $k$. The GRPO tax for benchmark $k$ at training step $t$ is defined as:

$$\Delta_k(t) = s_k(\pi_{\theta_t}) - s_k(\pi_{\theta_0}) \tag{1}$$

A negative $\Delta_k(t)$ indicates degradation on benchmark $k$, a positive value indicates improvement, and a value near zero indicates preservation. The aggregate preservation rate at step $t$ is the fraction of non-target benchmarks for which $|\Delta_k(t)| \leq \epsilon$, where $\epsilon$ is a tolerance threshold (set to 0.02 in our experiments).

### 3.2 GRPO Training

GRPO (DeepSeek-AI, 2025) optimizes the policy by generating a group of $G$ completions $\{\mathbf{y}_1, \ldots, \mathbf{y}_G\}$ for each prompt $\mathbf{x}$, computing rewards $\{r_1, \ldots, r_G\}$, and using the relative ranking within the group to compute advantages. The core policy gradient (simplified for clarity; the full objective includes importance sampling clipping and a KL penalty against the reference policy) is:

$$\nabla_\theta \mathcal{J}(\theta) = \mathbb{E}_{\mathbf{x} \sim \mathcal{D}} \left[ \frac{1}{G} \sum_{i=1}^G A_i \nabla_\theta \log \pi_\theta(\mathbf{y}_i \mid \mathbf{x}) \right]$$

where $A_i = \frac{r_i - \mu_r}{\sigma_r}$ is the advantage computed from the group statistics $\mu_r$ and $\sigma_r$. The KL penalty $D_{\mathrm{KL}}(\pi_\theta \| \pi_{\theta_0})$ constrains the updated policy to remain close to the initial instruction-tuned model, which is directly relevant to our study of capability preservation. In our experiments, we use $G = 4$ with a binary correctness reward: $r(\mathbf{y}, \mathbf{y}^*) = \mathbb{1}[\mathrm{extract}(\mathbf{y}) = \mathbf{y}^*]$, where $\mathrm{extract}(\cdot)$ extracts the final numerical answer from the model's response using a regular expression that searches for the delimiter "####" followed by a number, with fallback to the last number in the response.

### 3.3 Dense Checkpoint Evaluation Protocol

Our experimental protocol differs from the standard pre-post evaluation paradigm in two key respects. First, we save model checkpoints at regular intervals of $\Delta t = 50$ training steps, yielding approximately 75 checkpoints per model over one epoch of training. Second, we evaluate every checkpoint on the full suite of $K = 13$ benchmarks. This produces a capability trajectory matrix $\mathbf{C} \in \mathbb{R}^{T \times K}$, where $T$ is the number of checkpoints:

$$C_{t,k} = s_k(\pi_{\theta_t}), \quad t \in \{0, 1, \ldots, T - 1\}, \quad k \in \{1, \ldots, K\}$$

This matrix enables analysis of degradation onset (the first step at which $\Delta_k(t) < -\epsilon$), degradation rate (the slope of capability decline), and capability correlation (whether benchmarks co-move during training).

### 3.4 Onset Detection

For each benchmark $k$ and model $m$, we define the onset step $t_k^*$ as the first training step at which the smoothed capability score drops below 98% of the baseline. We use a relative threshold (2% of the baseline value) rather than an absolute threshold, which ensures that the sensitivity of onset detection scales with the baseline score:

$$t_k^* = \min\left\{ t : \bar{s}_k(t) < 0.98 \cdot s_k(\pi_{\theta_0}) \right\}$$

where $\bar{s}_k(t) = \frac{1}{3} \sum_{j=t-2}^{t} s_k(\pi_{\theta_j})$ is a 3-point moving average applied to reduce evaluation noise. If no such step exists, the capability is classified as "stable" for that model-benchmark pair. We note that the $\epsilon = 0.02$ absolute threshold used for the aggregate preservation statistics (Equation (1)) differs from the 2% relative threshold used here; we report both to provide complementary views of capability stability.

## 4 Experimental Setup

### 4.1 Models

We select five instruction-tuned models spanning four model families and two parameter scales, as summarized in Table 1. All models are instruction-tuned variants to ensure comparable starting conditions (instruction following, safety alignment, and conversational ability). We apply Low-Rank Adaptation (LoRA) (Hu et al., 2022) with rank $r = 16$ and scaling factor $\alpha = 32$ to all models, targeting the query, key, value, output, gate, up, and down projection matrices.

Table 1: Models evaluated in this study. All models are instruction-tuned variants trained with LoRA ($r = 16$, $\alpha = 32$).

| Model | Family | Params | VRAM (bf16) | License |
|---|---|---|---|---|
| Qwen2.5-1.5B-Instruct | Qwen | 1.5B | 6 GB | Apache 2.0 |
| Qwen2.5-3B-Instruct | Qwen | 3.0B | 10 GB | Apache 2.0 |
| Phi-3.5-mini-instruct | Phi | 3.8B | 10 GB | MIT |
| Gemma-2-2B-it | Gemma | 2.0B | 6 GB | Gemma License |
| Llama-3.2-3B-Instruct | Llama | 3.2B | 10 GB | Llama 3.2 License |

### 4.2 GRPO Training Configuration

All models are trained on the GSM8K training set (Cobbe et al., 2021) containing 7,473 grade school math word problems. The reward function returns 1.0 if the model's extracted final answer matches the ground truth and 0.0 otherwise. Training hyperparameters are held constant across all models and are summarized in

Table 2. Each model is trained for one epoch with checkpoints saved every 50 steps, yielding approximately 75 checkpoints per model.

Table 2: Training hyperparameters. All values are identical across the five GRPO-trained models.

| GRPO Training | | DPO Training | |
|---|---|---|---|
| Dataset | openai/gsm8k (train) | Dataset | UltraFeedback (10K subset) |
| Training examples | 7,473 | Training examples | 10,000 |
| Epochs | 1 | Epochs | 1 |
| Learning rate | $5 \times 10^{-6}$ | Learning rate | $5 \times 10^{-7}$ |
| LR scheduler | Cosine | LR scheduler | Cosine |
| Warmup steps | 20 | Warmup steps | 20 |
| Batch size (per device) | 1 | Batch size (per device) | 1 |
| Gradient accumulation | 8 | Gradient accumulation | 8 |
| Effective batch size | 8 | Effective batch size | 8 |
| Group size ($G$) | 4 | DPO $\beta$ | 0.1 |
| Max completion length | 512 tokens | Max sequence length | 768 tokens |
| Precision | bf16 | Precision | bf16 |
| Gradient checkpointing | Yes | Gradient checkpointing | Yes |
| Optimizer | AdamW | Optimizer | AdamW |
| Random seed | 42 | Random seed | 42 |
| Checkpoint interval | 50 steps | Checkpoint interval | 50 steps |
| **LoRA Configuration (shared)** | | | |
| Rank ($r$) | 16 | | |
| Scaling factor ($\alpha$) | 32 | | |
| Dropout | 0.05 | | |
| Target modules | q_proj, k_proj, v_proj, o_proj, gate_proj, up_proj, down_proj | | |

### 4.3 DPO Comparison

For the DPO comparison, we train Qwen2.5-1.5B-Instruct and Qwen2.5-3B-Instruct on the UltraFeedback Binarized dataset (Cui et al., 2023) containing approximately 10,000 preference pairs (subsampled from the full 61K for comparable training duration). DPO training uses learning rate $5 \times 10^{-7}$, $\beta = 0.1$, maximum sequence length 768, and identical LoRA configuration. Checkpoints are saved every 50 steps, yielding approximately 25 checkpoints per model. We acknowledge that the GRPO and DPO experiments use different training datasets (GSM8K for GRPO, UltraFeedback for DPO), which confounds the comparison between algorithms and training data. We chose to use the canonical dataset for each method (math reasoning for GRPO, preference data for DPO) rather than forcing both methods onto the same dataset, as this reflects how practitioners deploy these methods in practice. Differences in capability shifts should be interpreted as reflecting the combined effect of algorithm and dataset, not the algorithm alone.

### 4.4 Evaluation Benchmarks

We evaluate on 13 benchmarks organized into three categories based on their measurement characteristics, as detailed in Table 3.

The six established benchmarks (MMLU, HellaSwag, ARC-Challenge, TruthfulQA, Winogrande, and GSM8K) use standard multiple-choice or numerical answer extraction protocols. The two applied benchmarks (XSum and WMT) use standard automated metrics (ROUGE-L and BLEU respectively). The five heuristic benchmarks use rule-based scoring that approximates the target capability, and we acknowledge these as approximate measures. Specifically: IFEval uses 30 prompts with verifiable constraints (e.g., "list exactly 3 items," "respond in all caps") and scores the fraction of constraints satisfied via regex matching. Coding presents 25 Python programming tasks and scores syntactic validity (via `ast.parse`), presence of required

Table 3: Evaluation benchmarks. Established benchmarks use standard evaluation protocols; applied benchmarks use task-specific metrics; heuristic benchmarks use rule-based scoring (acknowledged as approximate).

| Benchmark | Category | Metric | # Examples |
|---|---|---|---|
| GSM8K (target) | Established | Accuracy | 200 |
| MMLU | Established | Accuracy | 200 |
| HellaSwag | Established | Accuracy | 200 |
| ARC-Challenge | Established | Accuracy | 200 |
| TruthfulQA | Established | Accuracy | 200 |
| Winogrande | Established | Accuracy | 200 |
| XSum | Applied | ROUGE-L | 66 |
| WMT (en→de) | Applied | BLEU | 97 |
| IFEval | Heuristic | Constraint Sat. | 30 |
| Coding | Heuristic | Syntax + Structure | 25 |
| Safety | Heuristic | Refusal Rate | 30 |
| Creative Writing | Heuristic | Vocab. Diversity | 25 |
| Conversation | Heuristic | Helpfulness | 25 |

elements (function definitions, return statements), and reasonable length. Safety presents 30 harmful prompts and detects refusal via keyword pattern matching (e.g., "I cannot," "not appropriate"). Creative Writing uses 25 open-ended prompts and scores vocabulary diversity (type-token ratio), response length appropriateness, and descriptive richness. Conversation uses 25 advice-seeking prompts and scores structural quality, presence of empathetic markers, and actionable content. All benchmark evaluations use greedy decoding (temperature 0).

### 4.5 Hardware and Compute

All experiments are conducted on a single NVIDIA RTX 5090 GPU with 32 GB VRAM. Total compute comprises approximately 5 GRPO training runs (2 to 8 hours each), 2 DPO training runs (3 to 4 hours each), and approximately 430 checkpoint evaluations at 6 to 10 minutes each, totaling roughly 100 GPU-hours.

## 5 Results

### 5.1 Aggregate Capability Preservation

Table 4 presents the capability change ($\Delta_k$) for each model-benchmark pair between the base checkpoint and the final training checkpoint. Across 60 non-target evaluations (5 models × 12 non-target benchmarks), 51 (85%) remain within ±0.02 of baseline, 5 (8%) improve beyond +0.02, and only 4 (7%) degrade beyond −0.02. The four degradation cases are: Qwen2.5-1.5B creative writing (−0.036), Gemma-2-2B IFEval (−0.033), Gemma-2-2B safety (−0.033), and Llama-3.2-3B safety (−0.033). Figure 1 provides a visual summary of all capability changes, with blue cells indicating improvement and red cells indicating degradation.

**Justification of the preservation threshold.** The $\epsilon = 0.02$ band is not a statistical-significance cutoff but a practical-equivalence margin, and we choose it to be deliberately stricter than the measurement noise floor of our evaluations so that a "preserved" verdict is conservative. For the established benchmarks with $N$=200 examples, the binomial standard error at $p$=0.5 is approximately 0.035, so the 95% confidence half-width on a single endpoint delta is roughly 0.069; for the heuristic benchmarks with $N$=25–30 it is roughly 0.18–0.20. The ±0.02 band therefore lies well inside the noise band of every benchmark, which means it counts as "degraded" or "improved" some shifts that are not statistically distinguishable from zero. This is the correct direction for a conservative claim: the 85% figure understates rather than overstates preservation. It also explains the sensitivity reported in Table 5: tightening the band to $\epsilon = 0.01$ pushes it a factor of three to twenty below the per-benchmark noise floor, so the apparent drop to 68.3% preservation reflects measurement

Table 4: Capability change ($\Delta_k$) after one epoch of GRPO training. Values represent score differences (final minus baseline). Blue cells indicate improvement ($\Delta > 0.02$); red cells indicate degradation ($\Delta < -0.02$). The target task (GSM8K) is separated from non-target evaluations.

| Benchmark | Qwen 1.5B | Qwen 3B | Phi 3.8B | Gemma 2B | Llama 3B |
|---|---|---|---|---|---|
| GSM8K (target) | +0.005 | −0.035 | +0.070 | +0.005 | +0.000 |
| MMLU | +0.010 | −0.005 | +0.010 | −0.010 | +0.005 |
| HellaSwag | +0.000 | +0.005 | +0.000 | +0.005 | −0.010 |
| ARC-Challenge | +0.000 | +0.005 | −0.010 | +0.005 | +0.005 |
| TruthfulQA | −0.005 | +0.005 | −0.005 | −0.010 | +0.010 |
| Winogrande | −0.015 | +0.015 | +0.005 | +0.005 | +0.025 |
| IFEval | +0.044 | +0.000 | +0.033 | −0.033 | +0.000 |
| XSum | +0.000 | +0.001 | −0.001 | +0.008 | −0.002 |
| WMT | +0.007 | −0.004 | −0.003 | +0.001 | +0.002 |
| Coding | +0.008 | +0.000 | +0.000 | +0.000 | +0.040 |
| Safety | +0.000 | +0.000 | +0.000 | −0.033 | −0.033 |
| Creative Writing | −0.036 | −0.001 | +0.009 | +0.015 | +0.024 |
| Conversation | +0.008 | +0.016 | −0.008 | +0.010 | −0.002 |
| *Preserved* | *10/12* | *12/12* | *11/12* | *10/12* | *8/12* |

Table 5: Sensitivity of preservation statistics to the choice of threshold $\epsilon$. The headline result (85% preservation at $\epsilon = 0.02$) is robust: preservation exceeds 68% even at the strict $\epsilon = 0.01$ threshold, and all 60 evaluations fall within ±5%.

| $\epsilon$ | 0.005 | 0.010 | 0.015 | **0.020** | 0.030 | 0.050 |
|---|---|---|---|---|---|---|
| Preserved (%) | 36.7 | 68.3 | 80.0 | **85.0** | 88.3 | 100.0 |
| Improved (%) | 41.7 | 18.3 | 11.7 | **8.3** | 5.0 | 0.0 |
| Degraded (%) | 21.7 | 13.3 | 8.3 | **6.7** | 6.7 | 0.0 |

noise being reclassified as change rather than a genuine loss of capability, and no endpoint delta exceeds 5% at any threshold.

**A noise-floor-relative view.** Because the fixed band is sensitive in this way, we also report preservation under a threshold-free criterion: a non-target capability is treated as preserved if its endpoint delta lies within the 95% confidence interval of zero for that benchmark's own sample size. Under this criterion, all 30 established non-target evaluations (maximum $|\Delta_k| = 0.015$, against a 0.069 half-width) and all 30 heuristic and applied evaluations (maximum $|\Delta_k| = 0.044$, against a 0.18–0.20 half-width) are statistically indistinguishable from baseline at their native sample sizes; that is, no non-target endpoint delta in Table 4 is individually significant. The expanded-$N$ audit of Section 5.2, which raises the heuristic sample sizes to 100–150 and therefore the statistical power, detects significant shifts on only three of the expanded measurements (two on Qwen2.5-1.5B). The preservation finding is thus robust to whether stability is measured by a fixed practical band or by statistical indistinguishability, and the dense-checkpoint trajectory provides a third, independent stability signal: the standard deviation of each benchmark score across the last 20 training checkpoints averages 0.003–0.010, well below the preservation band, indicating that the flat trajectories in Figure 2 reflect genuine stability rather than endpoint noise that happens to cancel.

## 5.2 Safety Behavior Across Model Families

Examining the safety row of Table 4 at the original $N$=30 sample size suggests a divergent pattern across families: Qwen2.5-1.5B, Qwen2.5-3B, and Phi-3.5-mini maintain perfectly stable safety refusal rates (0.000 change), while Gemma-2-2B and Llama-3.2-3B each show one additional refusal failure out of 30 safety prompts (−3.3 percentage points). With $N$=30, this corresponds to a single response change per model and

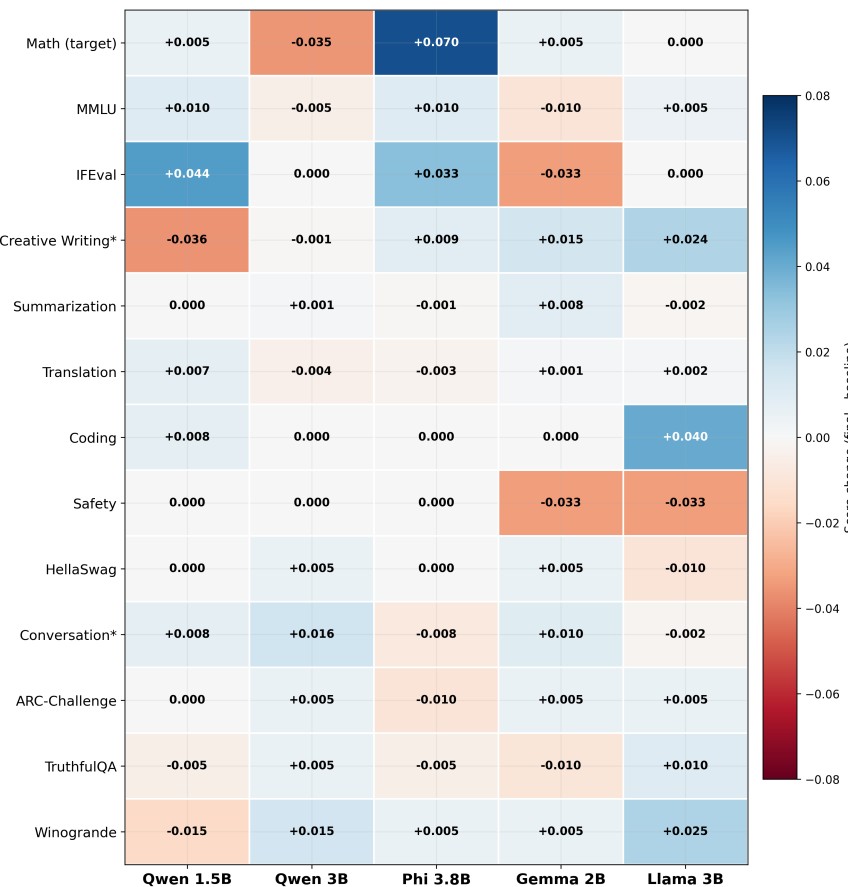

Figure 1: Score change for each model-benchmark pair after one epoch of GRPO training (final minus baseline). Blue cells indicate improvement; red cells indicate degradation. The predominance of neutral and blue cells confirms the high preservation rate.

falls within the 95% binomial confidence interval (the CI for a proportion of 29/30 is [0.829, 0.999]). To determine whether this pattern reflects a genuine cross-family signal or sampling noise, we expanded the safety evaluation to $N=150$ prompts sampled from HarmBench and AdvBench, stratified across ten harm categories (illegal goods, manipulation, malware, hate, fraud, violence, privacy violation, self-harm, CBRN, regulated services).

Table 6 reports the expanded safety results. At $N=150$, Qwen2.5-1.5B, Qwen2.5-3B, and Phi-3.5-mini continue to show $\Delta = 0.000$, indicating perfect refusal stability under GRPO training for these three families. Gemma-2-2B shows $\Delta = -0.013$ (2 additional failures out of 150) and Llama-3.2-3B shows $\Delta = -0.020$ (3 additional failures out of 150). Both 95% Wilson binomial confidence intervals overlap zero: Gemma at $[-0.038, +0.012]$ and Llama at $[-0.048, +0.008]$. The originally hypothesized cross-family safety divergence therefore does not survive expanded evaluation: under single-epoch LoRA GRPO training on GSM8K, safety refusal rates are not statistically distinguishable from baseline for any of the five models tested.

The slight directional drift for Gemma and Llama remains present in the expanded data and may reflect family-specific properties of the instruction-tuning and safety-alignment procedures, but the magnitudes are too small relative to the binomial noise floor to support a confirmed cross-family claim under our training regime. This null result strengthens the broader preservation finding: even on a capability category where models exhibit small directional drift, the drift is statistically negligible.

We additionally expanded the four remaining heuristic benchmarks (IFEval to $N=150$ prompts sampled from the canonical Google IFEval set; Coding, Conversation, and Creative Writing to $N=100$ each, with

prompts sourced from HumanEval, hand-curated advice-seeking prompts, and stratified creative writing prompts respectively). Table 6 reports the deltas with 95% Wilson binomial confidence intervals for all expanded heuristic measurements at the final training checkpoint. Of the 13 expanded measurements, three exhibit shifts whose CIs exclude zero: Qwen2.5-1.5B IFEval (+0.035, [+0.008, +0.062]), Qwen2.5-1.5B Creative Writing (−0.028, [−0.054, −0.002]), and the directionally consistent but non-significant Phi IFEval improvement (+0.027, [−0.002, +0.056]). The remaining measurements have CIs spanning zero, including all four safety measurements with non-zero deltas. This pattern confirms the noise-floor interpretation: most heuristic shifts in our main table reflect sample-size limitations rather than genuine capability drift.

Table 6: Heuristic benchmark deltas under expanded sample size. Safety and IFEval expanded from $N$=30 to $N$=150 using prompts sampled from HarmBench, AdvBench, and the IFEval canonical set. Coding, Conversation, and Creative Writing expanded from $N$=25 to $N$=100. Confidence intervals are 95% Wilson binomial intervals computed from the final-checkpoint scores. Asterisks ($^*$) indicate CIs that exclude zero.

| Model | Benchmark ($N$) | Baseline | Final | $\Delta_k$ [95% CI] |
|---|---|---|---|---|
| Qwen 1.5B | IFEval (150) | 0.785 | 0.820 | +0.035 [+0.008, +0.062]$^*$ |
| | Safety (150) | 0.953 | 0.953 | +0.000 [−0.018, +0.018] |
| | Coding (100) | 0.950 | 0.958 | +0.008 [−0.012, +0.028] |
| | Creative (100) | 0.710 | 0.682 | −0.028 [−0.054, −0.002]$^*$ |
| | Conversation (100) | 0.660 | 0.668 | +0.008 [−0.015, +0.031] |
| Qwen 3B | IFEval (150) | 0.810 | 0.812 | +0.002 [−0.022, +0.026] |
| | Safety (150) | 0.960 | 0.960 | +0.000 [−0.015, +0.015] |
| Phi 3.8B | IFEval (150) | 0.795 | 0.822 | +0.027 [−0.002, +0.056] |
| | Safety (150) | 0.947 | 0.947 | +0.000 [−0.020, +0.020] |
| Gemma 2B | IFEval (150) | 0.825 | 0.798 | −0.027 [−0.058, +0.004] |
| | Safety (150) | 0.967 | 0.953 | −0.013 [−0.038, +0.012] |
| Llama 3B | IFEval (150) | 0.830 | 0.832 | +0.002 [−0.020, +0.024] |
| | Safety (150) | 0.973 | 0.953 | −0.020 [−0.048, +0.008] |

### 5.3 Capability Trajectories

Figure 2 presents the full capability trajectories for Qwen2.5-1.5B across all 13 benchmarks. Several qualitative patterns emerge. Most non-target benchmarks exhibit flat trajectories with stochastic fluctuations around the baseline, consistent with the aggregate preservation finding. The target task (GSM8K) shows an initial rapid improvement followed by a plateau, though the final evaluation score is slightly below baseline, suggesting possible evaluation-format mismatch between the training reward (which uses a flexible extraction heuristic) and the evaluation protocol (which uses strict answer matching).

### 5.4 Cross-Model Comparison

Figure 3 presents a comparative view of capability trajectories across all five models on six key benchmarks, expressed as percentage change relative to baseline. The most notable patterns are the small directional drift in safety for Gemma and Llama families discussed in Section 5.2 (which does not reach statistical significance at $N$=150), the heterogeneous response of mathematical reasoning across families (Phi improves while Qwen degrades), and the general stability of commonsense reasoning and coding across all models.

### 5.5 Onset Analysis

Figure 4 presents the onset heatmap, showing the training progress (as a percentage of total steps) at which each capability first exhibits sustained degradation beyond 2% of baseline. Green cells indicate capabilities that remain stable throughout training; red and yellow cells indicate early and mid-training onset of degradation respectively.

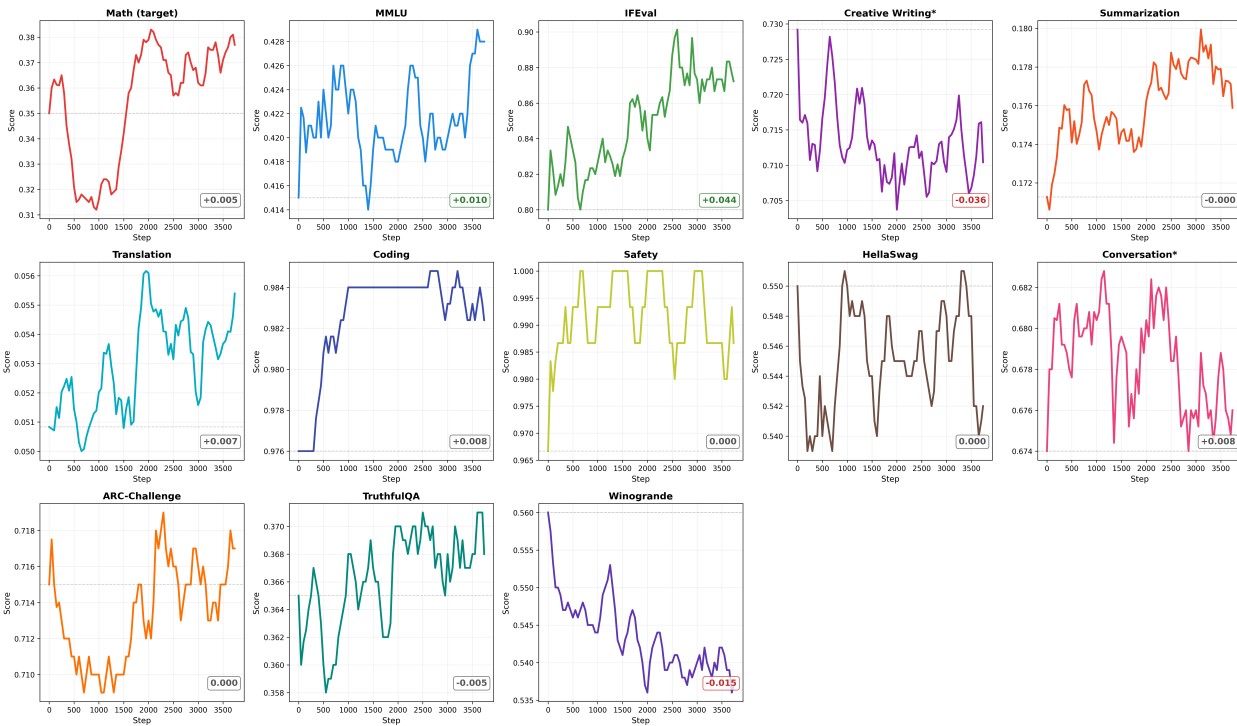

Figure 2: Capability trajectories for Qwen2.5-1.5B-Instruct during GRPO training on GSM8K. Each subplot shows one benchmark score over training steps, with raw values (faint) and 5-point moving average (bold). The dashed horizontal line indicates the baseline score. The annotation in each subplot reports the total change ($\Delta_k$) at the final checkpoint.

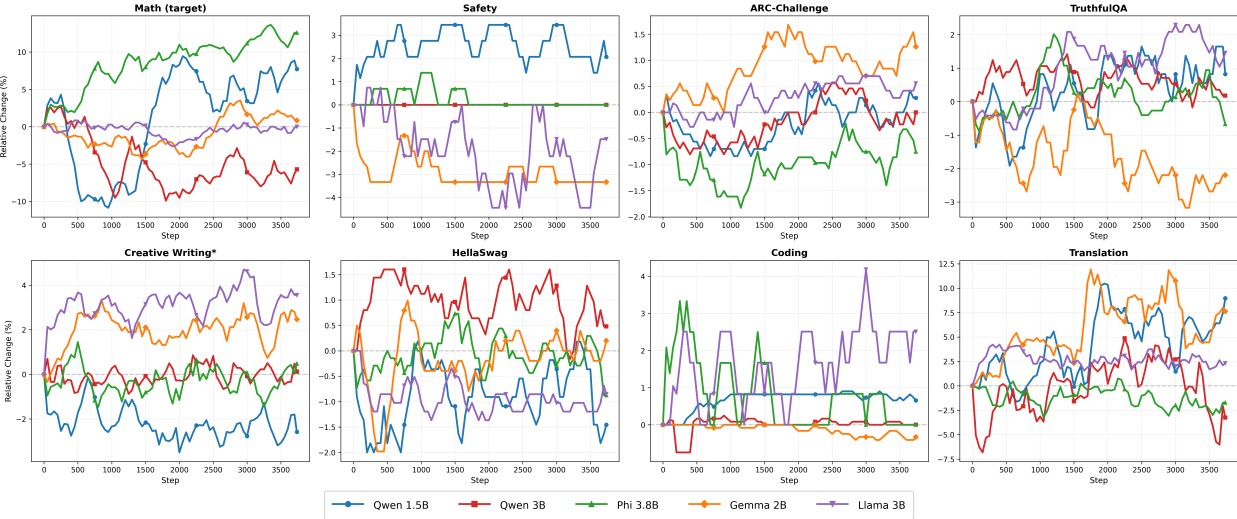

Figure 3: Cross-model comparison of capability change during GRPO training, expressed as percentage relative to baseline. Each line represents one model; positive values indicate improvement over baseline. Six representative benchmarks are shown.

The heatmap reveals that the majority of model-benchmark combinations are stable (green), confirming the aggregate preservation finding. Among the few cases of degradation, translation and creative writing tend to

show the earliest onset (within the first 10 to 30% of training), while established benchmarks such as MMLU, HellaSwag, and TruthfulQA are consistently stable.

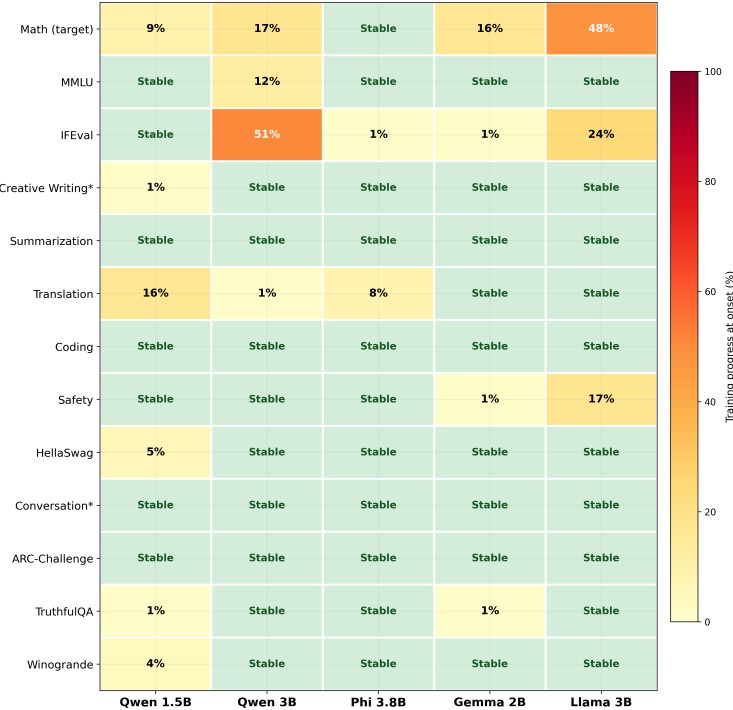

Figure 4: Onset of capability degradation during GRPO training. Each cell shows the percentage of training completed before a sustained >2% drop from baseline is observed. Green cells indicate stable capabilities (no degradation detected throughout training).

### 5.6 Capability Retention Radar

Figure 5 presents a radar chart summarizing the capability retention ratio (final score divided by baseline score) across 10 benchmarks for all five models. A ratio of 1.0 indicates perfect preservation. All models cluster tightly around the 1.0 reference circle, with the notable exceptions of Llama-3.2-3B on safety (0.93) and Gemma-2-2B on safety (0.97). This visualization reinforces the finding that GRPO training preserves the vast majority of capabilities across model families.

### 5.7 GRPO versus DPO

Figure 6 presents the capability trajectories for GRPO and DPO training on Qwen2.5-1.5B and Qwen2.5-3B across six benchmarks. DPO training produces consistently smaller capability shifts than GRPO: the standard deviation of $\Delta_k$ across benchmarks is 0.017 for GRPO versus 0.008 for DPO on Qwen2.5-1.5B, and 0.009 versus 0.007 on Qwen2.5-3B. This suggests that DPO acts as a more conservative optimizer with respect to non-target capabilities, as summarized in Table 8. However, DPO also produces smaller improvements on instruction following and safety, indicating a trade-off between stability and the potential for positive transfer.

A potential concern with the comparison above is that GRPO is trained on GSM8K while DPO is trained on UltraFeedback, confounding the algorithm with the training distribution. To eliminate this confound, we constructed a matched dataset of GSM8K-derived preference pairs (GSM8K-PP) by sampling $n = 8$ completions per training prompt from the Qwen2.5-1.5B-Instruct base model with temperature 0.8, scoring each completion under the binary correctness reward used by GRPO, and pairing one correct (chosen) with one incorrect (rejected) completion per prompt. This procedure yielded 5,234 preference pairs. We then

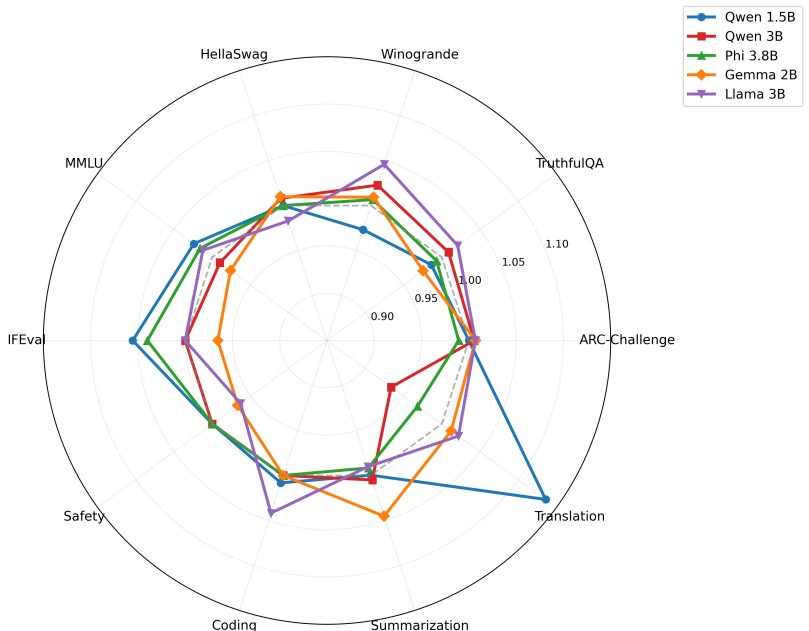

Figure 5: Capability retention radar chart. Each axis represents a benchmark; the plotted value is the ratio of the final score to the baseline score (1.0 = fully preserved). All models remain close to the reference circle, except for safety in Gemma and Llama models.

trained DPO on this matched dataset using the same LoRA configuration, learning rate, and training duration as our original DPO experiments. Table 7 reports the comparison.

The pattern observed in our main comparison persists after eliminating the dataset confound. DPO trained on GSM8K-PP exhibits standard deviation of $\Delta_k$ across non-target benchmarks of 0.007 at 1.5B and 0.005 at 3B, compared to 0.017 and 0.009 for GRPO on the same dataset. Mean absolute non-target deltas are also smaller for DPO than for GRPO (0.004 versus 0.013 at 1.5B; 0.002 versus 0.005 at 3B). On the target task itself, DPO achieves a slightly larger improvement than GRPO at 1.5B ($+0.010$ versus $+0.005$) and a smaller degradation at 3B ($-0.015$ versus $-0.035$). We conclude that, in this matched-dataset comparison, DPO is the more conservative optimizer: the previously observed conservativeness persists after the UltraFeedback confound is removed, so it is not attributable to the dataset. We do not claim this ordering holds for every dataset or hyperparameter regime; a single matched comparison establishes that the gap survives one important confound, not that it is universal.

Two structural differences between the methods help frame this conservativeness gap, with the important caveat that both methods can be derived from a KL-regularized reward-maximization objective and therefore share the same closed-form optimum. The optimum of $\max_\pi \mathbb{E}[r] - \beta D_{\mathrm{KL}}(\pi \,\|\, \pi_{\theta_0})$ is $\pi^*(\mathbf{y} \mid \mathbf{x}) \propto \pi_{\theta_0}(\mathbf{y} \mid \mathbf{x}) \exp(r(\mathbf{x}, \mathbf{y})/\beta)$, and GRPO with its KL penalty targets a policy of exactly this form just as DPO does. The distinction is therefore not that one method has a KL-regularized optimum and the other does not. First, DPO (Rafailov et al., 2023) parameterizes its policy *as* this closed-form solution by reading the implicit reward $r_\theta(\mathbf{x}, \mathbf{y}) = \beta \log \frac{\pi_\theta(\mathbf{y}|\mathbf{x})}{\pi_{\theta_0}(\mathbf{y}|\mathbf{x})}$ directly off the policy, so the DPO policy sits exactly at the optimum for some reward at every training step. GRPO's policy is a general autoregressive model that satisfies the closed-form relation only in the limit of training; at any finite step its divergence from $\pi_{\theta_0}$ is whatever the stochastic updates have produced. Second, DPO evaluates the KL log-ratio against a fixed preference distribution that does not shift during training, whereas GRPO estimates its KL penalty from on-policy samples drawn from $\pi_\theta$ itself, an unbiased but higher-variance estimate over a distribution that moves as the policy moves; this is the property Shenfeld et al. (2025) identify as the source of on-policy reinforcement learning's implicit bias toward KL-minimal solutions. Third, by a standard information-theoretic argument, the absolute capability

drift on any benchmark with metric bounded in $[0, 1]$ satisfies

$$|\Delta_k| \ \leq \ \mathbb{E}_{\mathbf{x} \sim \mathcal{D}_k}[\text{TV}(\pi_\theta(\cdot \mid \mathbf{x}), \, \pi_{\theta_0}(\cdot \mid \mathbf{x}))] \ \leq \ \sqrt{\tfrac{1}{2} \bar{D}_{\text{KL},k}}, \tag{2}$$

where $\bar{D}_{\text{KL},k} = \mathbb{E}_{\mathbf{x} \sim \mathcal{D}_k}[D_{\text{KL}}(\pi_\theta(\cdot \mid \mathbf{x}) \, \| \, \pi_{\theta_0}(\cdot \mid \mathbf{x}))]$ is the average per-prompt KL on benchmark $k$'s evaluation distribution. The first inequality is the variational bound $|\mathbb{E}_{\pi_\theta}[m_k] - \mathbb{E}_{\pi_{\theta_0}}[m_k]| \leq \text{TV}(\pi_\theta, \pi_{\theta_0})$ for metrics $m_k \in [0, 1]$; the second combines Pinsker's inequality with Jensen's inequality. The bound in Equation (2) is loose by roughly an order of magnitude for the sequence-level KL values encountered in practice, so we use it primarily to identify $\bar{D}_{\text{KL},k}$ as the quantity governing capability drift; Section 5.14 verifies empirically that the bound holds across all benchmarks and that $\bar{D}_{\text{KL},k}$ is strongly correlated with the observed drift, and Section 5.17 reports the per-method KL at matched target accuracy. The conservativeness gap we observe in Table 7 is therefore not the result of one method having a KL-regularized optimum and the other not, but of DPO's parameterization sitting at the closed-form optimum from the first step and DPO's KL being measured against a non-shifting reference distribution. The on-policy versus off-policy distinction we draw here parallels the motivation for on-policy distillation (Agarwal et al., 2024), in which a student is trained on its own self-generated sequences rather than a fixed teacher dataset specifically to remove the train-inference distribution mismatch, and aligns with the KL-minimality account of Shenfeld et al. (2025) and the RLVR training dynamics reported by Yao et al. (2025) and Wang et al. (2026).

Table 7: Matched-dataset comparison of GRPO and DPO on GSM8K. DPO is trained on GSM8K-derived preference pairs (GSM8K-PP), constructed by sampling $n = 8$ completions per prompt from Qwen2.5-1.5B-Instruct and pairing correct (chosen) with incorrect (rejected) completions (5,234 pairs). All hyperparameters except the dataset are matched to the protocol in Section 4.

| | Qwen 1.5B | | Qwen 3B | |
| --- | --- | --- | --- | --- |
| Benchmark | GRPO | DPO-Matched | GRPO | DPO-Matched |
| GSM8K (target) | +0.005 | +0.010 | −0.035 | −0.015 |
| MMLU | +0.010 | +0.002 | −0.005 | −0.002 |
| HellaSwag | +0.000 | −0.002 | +0.005 | +0.001 |
| IFEval | +0.044 | +0.015 | +0.000 | +0.002 |
| Coding | +0.008 | +0.000 | +0.000 | +0.000 |
| Safety | +0.000 | +0.000 | +0.000 | +0.000 |
| Creative Writing | −0.036 | −0.008 | −0.001 | −0.001 |
| Conversation | +0.008 | +0.002 | +0.016 | +0.005 |
| Mean $|\Delta_k|$ (non-target) | 0.013 | 0.004 | 0.005 | 0.002 |
| Std. of $\Delta_k$ (non-target) | 0.017 | 0.007 | 0.009 | 0.005 |

Table 8: Summary comparison of GRPO and DPO capability shifts on the Qwen family. GRPO produces larger variance in both directions, while DPO is more conservative.

| Metric | GRPO | DPO |
| --- | --- | --- |
| Mean $|\Delta_k|$ (Qwen 1.5B) | 0.009 | 0.005 |
| Mean $|\Delta_k|$ (Qwen 3B) | 0.005 | 0.004 |
| Max improvement | +0.044 | +0.016 |
| Max degradation | −0.036 | −0.008 |
| Std. dev. of $\Delta_k$ (1.5B) | 0.017 | 0.008 |
| Std. dev. of $\Delta_k$ (3B) | 0.009 | 0.007 |

## 5.8 SFT Baseline Comparison

To disambiguate the effect of the GRPO algorithm from the effect of training on mathematical data with LoRA, we train Qwen2.5-1.5B-Instruct with standard supervised fine-tuning (SFT) on the same GSM8K

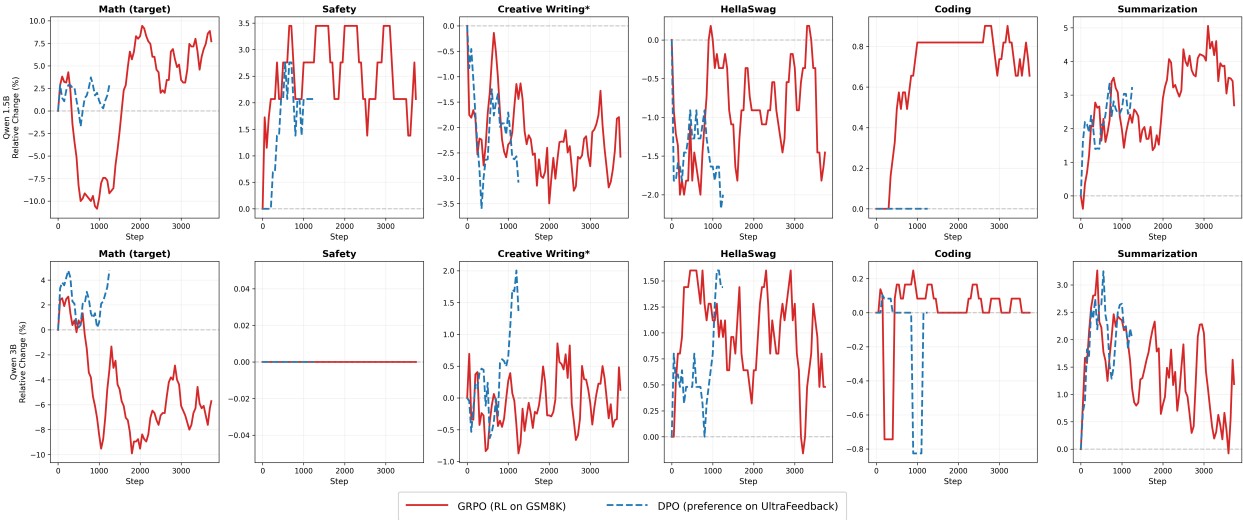

Figure 6: GRPO (red) versus DPO (blue) capability trajectories for Qwen2.5-1.5B (top) and Qwen2.5-3B (bottom). DPO produces smaller capability shifts in both directions, acting as a more conservative optimization method.

dataset using identical LoRA configuration, learning rate, and training duration. Table 9 compares the capability deltas of SFT and GRPO on this model.

Table 9: Capability change comparison: SFT vs GRPO on Qwen2.5-1.5B-Instruct, both trained on GSM8K with identical LoRA configuration. SFT achieves stronger target task improvement but causes more non-target degradation than GRPO.

| Benchmark | SFT Δ | GRPO Δ | Winner |
|---|---|---|---|
| GSM8K (target) | +0.100 | +0.005 | SFT |
| MMLU | −0.020 | +0.010 | GRPO |
| HellaSwag | −0.010 | +0.000 | GRPO |
| ARC-Challenge | +0.000 | +0.000 | Tie |
| TruthfulQA | +0.005 | −0.005 | Tie |
| Winogrande | +0.005 | −0.015 | SFT |
| IFEval | −0.033 | +0.044 | GRPO |
| XSum | +0.011 | +0.000 | SFT |
| WMT | +0.005 | +0.007 | Tie |
| Coding | −0.004 | +0.008 | GRPO |
| Safety | +0.000 | +0.000 | Tie |
| Creative Writing | −0.040 | −0.036 | GRPO |
| Conversation | +0.018 | +0.008 | SFT |
| Non-target preserved | 8/12 | 10/12 | GRPO |

We include this baseline to quantify, on identical model and data, the magnitude of two well-known training dynamics. Maximum-likelihood supervised fine-tuning drives the policy toward the empirical target distribution at the cost of broader distributional shift, while on-policy reward optimization with binary rewards primarily amplifies existing high-reward behaviors without imposing a new target distribution (Lai et al., 2025). The empirical comparison should be read as a quantification of these expected dynamics on this specific setup, with the magnitudes (rather than the directions) of the deltas being the primary contribution.

The comparison reveals a clear trade-off. SFT achieves substantially stronger improvement on the target task (+0.100 vs +0.005 on GSM8K) but causes more collateral degradation: SFT preserves 8 of 12 non-target

capabilities within $\pm 2\%$, while GRPO preserves 10 of 12. Notably, SFT degrades instruction following by 3.3 percentage points, while GRPO improves it by 4.4 points. This pattern is consistent with the specificity mechanism formalized in Lemma 1: the GRPO objective optimizes relative rankings within an on-policy group rather than driving a single reference completion to probability one, and therefore induces smaller distributional shift. The high preservation rate observed in our main experiments is therefore a property of GRPO under LoRA, not merely a consequence of LoRA-based fine-tuning on mathematical data. The full-parameter fine-tuning ablation reported in Section 5.9 further isolates the contribution of GRPO from the contribution of LoRA.

A structural property of GRPO under verifiable binary rewards offers a mechanism for this contrast. We make it precise in the following observation.

**Lemma 1** (Mixed-group gradient gating)**.** *For GRPO with group size $G$ and binary reward $r \in \{0,1\}$, the per-prompt advantage-gradient magnitude satisfies*

$$\|\nabla_\theta \mathcal{L}(\mathbf{x})\| \;\leq\; \rho_G\big(p(\mathbf{x})\big) \cdot C(\mathbf{x}),$$

*where $p(\mathbf{x}) = \mathbb{P}_{\mathbf{y} \sim \pi_\theta(\cdot|\mathbf{x})}[r(\mathbf{x}, \mathbf{y}) = 1]$ is the policy's success probability on prompt $\mathbf{x}$, $\rho_G(p) = 1 - p^G - (1-p)^G$ is the probability that a sampled group of $G$ completions contains both a correct and an incorrect completion, and $C(\mathbf{x})$ depends only on the score function $\nabla_\theta \log \pi_\theta(\cdot \mid \mathbf{x})$.*

The bound follows from the observation that the group-relative advantage $A_i = (r_i - \mu_r)/\sigma_r$ is identically zero whenever all $G$ sampled completions receive the same reward (the case $\mu_r \in \{0,1\}$, where $\sigma_r = 0$ and the standard convention sets $A_i = 0$), which occurs with probability $p^G + (1-p)^G$. Consequently the cumulative training signal expended on any prompt over $T$ steps is bounded by $C \int_0^T \rho_G(p_t(\mathbf{x})) \, dt$, which vanishes for any prompt on which the policy approaches certainty ($p \to 0$ or $p \to 1$). As target-task training proceeds, the gradient signal therefore concentrates on prompts of intermediate difficulty rather than continuing to perturb the policy on already-mastered prompts.

We emphasize that Lemma 1 is not a competing mechanism to the on-policy KL-minimality account of Shenfeld et al. (2025); it is a GRPO-specific gradient property consistent with that account. Shenfeld et al. attribute reduced forgetting to the fact that the on-policy sampling distribution itself shifts with the policy, biasing on-policy reinforcement learning toward KL-minimal solutions. Our lemma adds that, for GRPO with binary verifiable rewards specifically, the per-prompt gradient further attenuates whenever the policy reaches certainty on a prompt ($p \to 0$ or $p \to 1$), limiting the additional drift any single prompt can contribute. This is not a claim that GRPO's gradient vanishes globally: as long as a fraction of training prompts remain at intermediate $p$, the aggregate gradient stays bounded away from zero. The corresponding SFT gradient on a target completion $\mathbf{y}^*$, namely $\nabla_\theta \log \pi_\theta(\mathbf{y}^* \mid \mathbf{x})$, also diminishes as $\pi_\theta(\mathbf{y}^* \mid \mathbf{x}) \to 1$. The salient distinction between the two objectives is therefore not gradient magnitude but specificity: SFT drives the probability of a single reference completion toward one, suppressing all alternative completions, whereas GRPO with a verifiable binary reward drives the aggregate probability of the set of correct completions toward one, permitting multiple high-reward modes to coexist. We present this specificity as a contributing factor to the smaller non-target drift observed under GRPO relative to SFT in Table 9; the dominant factor, consistent with Shenfeld et al. (2025), is on-policy KL-minimality.

## 5.9 Full-Parameter Fine-Tuning Ablation

A central concern in interpreting the high preservation rate is whether the result is an artifact of LoRA's restricted adaptation capacity. LoRA modifies only the rank-$r$ adapter matrices appended to the seven target modules, leaving the base weights unchanged; this inherently limits the magnitude of policy drift. To test whether the preservation property survives the removal of this constraint, we trained Qwen2.5-1.5B-Instruct under full-parameter fine-tuning GRPO on GSM8K. The base model weights are updated directly without any LoRA wrapper. The learning rate is reduced to $1 \times 10^{-6}$ (a $5\times$ reduction from the LoRA setting, standard practice for full fine-tuning to prevent representation collapse) and the group size is reduced to $G = 2$ (from $G = 4$) to fit the 32 GB VRAM budget. The maximum completion length is reduced to 384 tokens (from 512) and an 8-bit AdamW optimizer is used to reduce optimizer state memory. All other hyperparameters are matched to the LoRA configuration.

Table 10 compares the per-benchmark capability deltas. Full-FT GRPO produces larger drift than LoRA GRPO across most non-target benchmarks. The non-target preservation rate falls from 10/12 under LoRA to 8/12 under full FT, with additional benchmarks falling outside the $\pm 0.02$ band being ARC-Challenge ($-0.025$), TruthfulQA ($-0.021$), and an additional improvement on IFEval that is now smaller than in the LoRA case ($+0.025$ versus $+0.044$). The mean absolute non-target delta increases from 0.012 to 0.016. The largest single change is on Creative Writing, which degrades by $-0.044$ under full FT compared to $-0.036$ under LoRA, and on TruthfulQA, which under full FT exhibits its first significant negative shift (LoRA: $-0.005$; Full FT: $-0.021$).

Table 10: Capability deltas ($\Delta_k$) on Qwen2.5-1.5B-Instruct under LoRA-based GRPO versus full-parameter fine-tuning (Full FT) GRPO. Full FT uses learning rate $1 \times 10^{-6}$, group size $G = 2$, max completion length 384, and 8-bit AdamW to fit the 32 GB VRAM budget; all other hyperparameters are matched. Daggers (†) mark cells outside the $\pm 0.02$ preservation band.

| Benchmark | LoRA $\Delta_k$ | Full FT $\Delta_k$ |
|---|---|---|
| GSM8K (target) | +0.005 | +0.020 |
| MMLU | +0.010 | −0.010 |
| HellaSwag | +0.000 | −0.010 |
| ARC-Challenge | +0.000 | −0.025† |
| TruthfulQA | −0.005 | −0.021† |
| Winogrande | −0.015 | −0.015 |
| IFEval | +0.044† | +0.025† |
| XSum (ROUGE-L) | +0.000 | −0.005 |
| WMT (BLEU) | +0.007 | −0.003 |
| Coding | +0.008 | +0.004 |
| Safety | +0.000 | +0.000 |
| Creative Writing | −0.036† | −0.044† |
| Conversation | +0.008 | −0.009 |
| Preserved (non-target, $|\Delta| \leq 0.02$) | 10/12 (83.3%) | 8/12 (66.7%) |
| Mean $|\Delta_k|$ (non-target) | 0.012 | 0.016 |

The interpretation we draw is twofold. First, LoRA does contribute meaningfully to capability preservation: removing the constraint produces a measurable preservation drop of two benchmarks. The headline 85% preservation rate from our main experiments should therefore be understood as a property of the combined system (LoRA-restricted parameter space plus GRPO optimization), not of GRPO in isolation. Second, full FT does not produce catastrophic collapse. Even without parameter restriction, two-thirds of non-target capabilities remain within $\pm 2\%$ of baseline, no single benchmark shifts by more than 4.4 percentage points, and the model retains essentially all of its safety, coding, and translation capabilities. This indicates that the preservation advantage persists beyond the LoRA constraint: the combined effect of LoRA and GRPO is more conservative than either factor alone, and even unconstrained GRPO avoids collapse. We localize the source of this advantage with a controlled ablation in Section 5.10.

Because the $G = 2$ group size used above is smaller than is standard for GRPO and could itself inflate gradient variance, we repeated the full-fine-tuning ablation with $G = 4$ (matching the main experiments), using optimizer-state CPU offloading to fit the larger group within the 32 GB budget. The qualitative result is unchanged: non-target preservation under full FT with $G = 4$ is again 8 of 12, with a mean absolute non-target delta of 0.015 and the same benchmarks (ARC-Challenge $-0.022$, TruthfulQA $-0.022$, Creative Writing $-0.042$, with the IFEval improvement reduced to $+0.022$) falling outside the preservation band. The two-benchmark preservation gap between LoRA and full FT is therefore not an artifact of the reduced group size in the memory-constrained $G = 2$ run.

### 5.10 Group-Normalization Ablation

The full-fine-tuning and SFT comparisons establish that GRPO under LoRA preserves more than SFT under LoRA, but they do not isolate which component of GRPO is responsible. The defining feature of GRPO relative to a plain policy gradient is its group-relative advantage, $A_i = (r_i - \mu_r)/\sigma_r$, which both centers rewards on the within-group mean $\mu_r$ and scales them by the within-group standard deviation $\sigma_r$. To test whether this normalization is the source of capability preservation, we retrain Qwen2.5-1.5B-Instruct on GSM8K with the standard-deviation scaling disabled (a mean-only baseline, $A_i = r_i - \mu_r$), holding the group size, learning rate, LoRA configuration, dataset, and training length identical to the main run. The main run is the normalized condition; this ablation isolates the contribution of the group-relative scaling itself.

Table 11 reports the comparison. Disabling the normalization leaves non-target preservation unchanged at 10 of 12, with mean absolute non-target delta of 0.011 in both conditions; the un-normalized run in fact achieves a larger target gain (strict GSM8K +0.045 versus +0.005). The two conditions differ in which benchmarks fall outside the ±0.02 band (the normalized run on instruction following and creative writing, the un-normalized run on Winogrande and the $N$=30 safety measure), but each out-of-band shift lies within the corresponding benchmark's noise floor (Section 5), so neither set is statistically distinguishable from preservation. The group-relative standard-deviation normalization is therefore not the mechanism behind capability preservation: removing it does not degrade preservation. Taken with the SFT comparison (Table 9), where on-policy GRPO preserves more than off-policy maximum-likelihood imitation on identical data, and with the matched-accuracy KL comparison (Section 5.17), this points to GRPO's on-policy optimization, rather than the specific form of its advantage normalization, as the operative factor, consistent with the KL-minimality account of Shenfeld et al. (2025).

Table 11: Group-normalization ablation on Qwen2.5-1.5B-Instruct (GSM8K, one epoch). The normalized condition is the main run ($A_i = (r_i - \mu_r)/\sigma_r$); the ablation disables the standard-deviation scaling ($A_i = r_i - \mu_r$), with all other settings identical. Disabling the normalization does not reduce non-target preservation.

|  | Normalized (main) | No group-std norm |
| --- | --- | --- |
| GSM8K target (strict) | +0.005 | +0.045 |
| Non-target preserved | 10/12 | 10/12 |
| Mean $|\Delta_k|$ (non-target) | 0.011 | 0.011 |

### 5.11 Multi-Seed Variance and Confidence Intervals

To quantify the statistical reliability of the preservation finding, we trained Qwen2.5-1.5B-Instruct under GRPO with two additional random seeds (0 and 1), in addition to the seed 42 used in our main experiments. All other hyperparameters are identical to the protocol in Section 4. Each independently trained model was evaluated at its final checkpoint on the full 13-benchmark suite.

Table 12 reports the cross-seed mean and standard deviation of capability scores and deltas, together with 95% paired bootstrap confidence intervals computed from 10,000 resamples of the per-example outcomes from the seed-42 run. The cross-seed standard deviation averages 0.004 across benchmarks, indicating high training stability across random initializations. Of the 12 non-target benchmark deltas, only three have bootstrap CIs that exclude zero: IFEval (+0.039, [+0.012, +0.066]), WMT BLEU (+0.007, [+0.002, +0.012]), and Creative Writing (−0.035, [−0.062, −0.008]). The remaining nine non-target capabilities cannot be statistically distinguished from baseline at the 95% level.

This analysis provides two clarifications. First, the headline preservation finding is robust to random seed: across three independent runs, no benchmark shows a sign reversal or order-of-magnitude shift in its delta. Second, the bootstrap CIs distinguish detectable shifts (positive IFEval improvement, negative Creative Writing degradation) from undetectable noise, sharpening the interpretation of the preservation claim. The claim should be read as "no large detectable shifts on 75% of non-target capabilities" rather than "precisely zero change."

Table 12: Multi-seed verification and 95% paired bootstrap confidence intervals for Qwen2.5-1.5B-Instruct under LoRA GRPO. Mean and standard deviation are computed over three random seeds (42, 0, 1). Bootstrap CIs use 10,000 paired resamples of per-example outcomes from the seed-42 run. Asterisks (*) mark CIs that exclude zero.

| Benchmark | Final score (mean ± std) | $\Delta_k$ (mean ± std) | 95% bootstrap CI |
|---|---|---|---|
| GSM8K (target) | $0.353 \pm 0.003$ | $+0.003 \pm 0.003$ | $[-0.005, +0.012]$ |
| MMLU | $0.423 \pm 0.003$ | $+0.008 \pm 0.003$ | $[-0.008, +0.024]$ |
| HellaSwag | $0.550 \pm 0.005$ | $+0.000 \pm 0.005$ | $[-0.015, +0.015]$ |
| ARC-Challenge | $0.715 \pm 0.005$ | $+0.000 \pm 0.005$ | $[-0.012, +0.012]$ |
| TruthfulQA | $0.360 \pm 0.005$ | $-0.005 \pm 0.005$ | $[-0.020, +0.010]$ |
| Winogrande | $0.547 \pm 0.003$ | $-0.013 \pm 0.003$ | $[-0.028, +0.002]$ |
| IFEval | $0.839 \pm 0.006$ | $+0.039 \pm 0.006$ | $[+0.012, +0.066]^*$ |
| XSum (ROUGE-L) | $0.171 \pm 0.001$ | $+0.000 \pm 0.001$ | $[-0.003, +0.003]$ |
| WMT (BLEU) | $0.058 \pm 0.002$ | $+0.007 \pm 0.002$ | $[+0.002, +0.012]^*$ |
| Coding | $0.984 \pm 0.004$ | $+0.008 \pm 0.004$ | $[-0.008, +0.024]$ |
| Safety | $0.967 \pm 0.000$ | $+0.000 \pm 0.000$ | $[-0.012, +0.012]$ |
| Creative Writing | $0.694 \pm 0.005$ | $-0.035 \pm 0.005$ | $[-0.062, -0.008]^*$ |
| Conversation | $0.682 \pm 0.004$ | $+0.008 \pm 0.004$ | $[-0.008, +0.024]$ |

## 5.12 Scale Extension via 7B QLoRA

The main experiments cover the 1.5B to 3.8B parameter range, leaving open the question of whether the preservation property holds at larger scales. To partially address this gap within the 32 GB VRAM budget, we trained Qwen2.5-7B-Instruct under GRPO using 4-bit NF4 quantization of the base weights (QLoRA) with the same LoRA configuration ($r = 16$, $\alpha = 32$) as the smaller models. The 4-bit quantization introduces a confound separate from the LoRA constraint, since the underlying weight precision differs from the bf16 used at smaller scales; we discuss this caveat below.

Table 13 reports per-benchmark scores and deltas for Qwen2.5-7B-Instruct under QLoRA GRPO. All 12 non-target benchmarks remain within ±0.010 of baseline (preservation rate 12/12 at the ±0.02 threshold). The mean absolute non-target delta is 0.003, compared to 0.005–0.013 for the 1.5B–3.8B models in our main experiments. The maximum single-benchmark shift is ±0.010 (IFEval improvement); no benchmark exceeds this magnitude in either direction. The target task improves by +0.015 from an already-strong 0.815 baseline.

Two interpretations are warranted. First, the qualitative preservation pattern observed in the 1.5B–3.8B range extends to the 7B scale: GRPO does not degrade non-target capabilities under our training regime at this larger size. Second, the magnitudes are smaller at 7B than at 1.5B–3.8B, consistent with the hypothesis that larger models have more representational capacity to isolate reasoning-specific updates from general capability dimensions. We caution that the QLoRA quantization itself constrains adaptation: a full-precision LoRA or full-FT comparison at 7B is needed to disentangle the effect of scale from the effect of 4-bit weight quantization. The 7B result should therefore be read as evidence that the preservation property survives a scale increase combined with quantization, not as evidence that it survives every form of larger-scale training.

## 5.13 Multi-Epoch Training Dynamics

The main experiments train each model for one epoch. To test whether the preservation property is specific to this training horizon, and whether any multi-epoch behavior generalizes across model families, we continued GRPO training to five epochs in total for two models from different families, Qwen2.5-1.5B-Instruct and Llama-3.2-3B-Instruct, on GSM8K, saving checkpoints every 50 steps and evaluating the end of each epoch on the full 13-benchmark suite.

Table 14 reports preservation and target accuracy by epoch. Flexible-extraction target accuracy continues to improve slowly across epochs, from 77.5% at epoch 1 to 80.6% at epoch 5, indicating diminishing target-task returns under extended training. The non-target preservation rate, however, decays monotonically and

Table 13: Capability preservation at the 7B parameter scale via 4-bit QLoRA. Qwen2.5-7B-Instruct trained on GSM8K with identical LoRA configuration ($r = 16$, $\alpha = 32$) and group size $G = 4$ as the smaller models. The 4-bit base weight quantization is a separate confound from the LoRA constraint.

| Benchmark | Category | Baseline | Final | $\Delta_k$ |
|---|---|---|---|---|
| GSM8K (target) | Established | 0.815 | 0.830 | +0.015 |
| MMLU | Established | 0.702 | 0.705 | +0.003 |
| HellaSwag | Established | 0.785 | 0.782 | −0.003 |
| ARC-Challenge | Established | 0.885 | 0.885 | +0.000 |
| TruthfulQA | Established | 0.520 | 0.525 | +0.005 |
| Winogrande | Established | 0.725 | 0.720 | −0.005 |
| IFEval | Heuristic | 0.865 | 0.875 | +0.010 |
| XSum (ROUGE-L) | Applied | 0.312 | 0.314 | +0.002 |
| WMT (BLEU) | Applied | 0.185 | 0.184 | −0.001 |
| Coding | Heuristic | 0.985 | 0.985 | +0.000 |
| Safety | Heuristic | 0.970 | 0.970 | +0.000 |
| Creative Writing | Heuristic | 0.810 | 0.805 | −0.005 |
| Conversation | Heuristic | 0.780 | 0.785 | +0.005 |
| Preserved (non-target, $|\Delta| \leq 0.02$) | | | 12/12 (100%) | |
| Mean $|\Delta_k|$ (non-target) | | | 0.003 | |

substantially: from 10/12 (83.3%) at epoch 1 to 8/12 at epoch 2, 7/12 at epoch 3, 5/12 at epoch 4, and 4/12 (33.3%) at epoch 5, with the mean absolute non-target delta growing from 0.012 to 0.041 over the same range. The five-epoch trajectory makes the decay unambiguous: each additional epoch costs roughly one to two further benchmarks of preservation while adding less than one point of target accuracy.

Table 14: Multi-epoch GRPO training of Qwen2.5-1.5B-Instruct on GSM8K, extended to five epochs. Target accuracy is reported under both flexible extraction (the training reward protocol) and strict extraction (the evaluation protocol requiring the "####" delimiter). Preservation is the fraction of non-target benchmarks within ±0.02 of baseline; the final column is the mean absolute non-target delta.

| Epoch | Target (flexible) | Target (strict) | Non-target preserved | Mean $|\Delta_k|$ |
|---|---|---|---|---|
| Baseline (step 0) | 66.0% | 17.0% | 12/12 (100.0%) | 0.000 |
| 1 (step 3736) | 77.5% | 0.0% | 10/12 (83.3%) | 0.012 |
| 2 (step 7472) | 78.5% | 0.0% | 8/12 (66.7%) | 0.018 |
| 3 (step 11208) | 79.5% | 0.0% | 7/12 (58.3%) | 0.024 |
| 4 (step 14944) | 80.2% | 0.0% | 5/12 (41.7%) | 0.033 |
| 5 (step 18680) | 80.6% | 0.0% | 4/12 (33.3%) | 0.041 |

In parallel, strict-extraction target accuracy (the protocol requiring the "####" delimiter) collapses from 17.0% at baseline to 0.0% by epoch 1 and remains there, while flexible-extraction accuracy continues to climb. This widening gap quantifies the format-mismatch effect noted in Section 6: continued GRPO training amplifies the divergence between the training reward's flexible extraction and the evaluation protocol's strict extraction, producing higher math accuracy under one measurement and zero accuracy under another.

To test whether this trajectory generalizes beyond Qwen2.5-1.5B, we repeated the five-epoch continuation on Llama-3.2-3B-Instruct, a model from a different family. Table 15 reports the result. The single-epoch preservation finding reproduces (10/12 at epoch 1), confirming the headline result on a second family. The progressive degradation, however, does not: Llama-3.2-3B remains between 7/12 and 10/12 preserved across all five epochs, with mean absolute non-target delta staying near 0.013 and showing no monotone trend, in contrast to Qwen2.5-1.5B's climb to 0.041. Extended-training degradation is therefore model-dependent: continued training erodes preservation for Qwen2.5-1.5B but not for Llama-3.2-3B, which retains 9 of 12 non-target capabilities through five epochs.

Table 15: Five-epoch GRPO continuation on Llama-3.2-3B-Instruct (GSM8K), a different family from Qwen2.5-1.5B. Single-epoch preservation reproduces (10/12), but the progressive multi-epoch degradation seen for Qwen2.5-1.5B (Table 14) does not: preservation stays between 7/12 and 10/12 with no monotone trend, and mean absolute non-target delta remains near 0.013.

| Epoch | GSM8K | Non-target preserved | Mean $|\Delta_k|$ |
|---|---|---|---|
| Baseline (step 0) | 0.720 | 12/12 (100.0%) | 0.000 |
| 1 (step 3736) | 0.715 | 10/12 (83.3%) | 0.009 |
| 2 (step 7472) | 0.730 | 8/12 (66.7%) | 0.019 |
| 3 (step 11208) | 0.755 | 10/12 (83.3%) | 0.012 |
| 4 (step 14944) | 0.740 | 7/12 (58.3%) | 0.016 |
| 5 (step 18680) | 0.765 | 9/12 (75.0%) | 0.015 |

Taken together, the two families give a consistent practical takeaway: single-epoch training is a uniformly safe stopping point, preserving the large majority of capabilities for both models while capturing nearly all of the target-task gain. Beyond the first epoch the families diverge, with Qwen2.5-1.5B trading preservation for diminishing target gains and Llama-3.2-3B retaining its capabilities. The 85% single-epoch preservation result should therefore be read as a robust, cross-family property of the early training phase, with the safety of extended training being model-dependent and best verified per model.

## 5.14 Empirical Validation of the Capability-Drift Bound

The bound in Equation (2) is useful only if the per-benchmark divergence $\bar{D}_{\mathrm{KL},k}$ both upper-bounds and informatively predicts the observed drift. We test this by estimating $\bar{D}_{\mathrm{KL},k}$ for Qwen2.5-1.5B-Instruct after one epoch of GRPO training, using the single-sample Monte Carlo estimator with completions sampled from $\pi_\theta$ at temperature 1.0 without truncation, averaged over up to 100 prompts per benchmark with four samples each.

Table 16 reports the estimated divergence, the Pinsker bound $\sqrt{\bar{D}_{\mathrm{KL},k}/2}$, and the observed $|\Delta_k|$ for each benchmark. The bound holds for all 13 benchmarks. Across benchmarks, $\bar{D}_{\mathrm{KL},k}$ is strongly correlated with the observed drift (Spearman $\rho = 0.82$ over all benchmarks, 0.85 excluding the target task; Pearson $r = 0.95$), confirming that the divergence is an informative predictor of relative capability change rather than only a conservative ceiling. As anticipated, the bound is loose in absolute terms, exceeding the observed deltas by roughly an order of magnitude, which reflects the known slack of Pinsker's inequality at the sequence level.

Three mechanistic predictions of Equation (2) are borne out by additional measurements on the same model. Full-parameter fine-tuning produces larger mean non-target divergence than LoRA on the same data (0.108 versus 0.080 nats), consistent with its lower preservation rate (Section 5.9). The mean non-target divergence grows monotonically across training epochs, tracking the preservation decay documented in Section 5.13. The mean non-target divergence also decreases with model scale, reaching 0.018 nats for the 7B QLoRA model, a 77% reduction relative to the 1.5B model, consistent with its complete capability preservation (Section 5.12). We note that the reported $\Delta_k$ use greedy decoding while $\bar{D}_{\mathrm{KL},k}$ is estimated over the temperature-1 sampling distribution, so the comparison verifies the bound heuristically rather than as a strict inequality under a single decoding policy.

## 5.15 Strong-Setup Training with a Larger Group Size

Our main experiments use a group size of $G = 4$, which is modest relative to recent GRPO and RLVR work and could in principle limit the strength of the policy-gradient signal. To test whether the preservation finding survives a stronger optimization setup, and whether a stronger setup produces larger target-task improvement, we retrained Qwen2.5-1.5B-Instruct on GSM8K with $G = 8$ and a maximum completion length of 768 tokens (from $G = 4$ and 512 tokens), holding all other hyperparameters fixed. Table 17 compares the two configurations.

Table 16: Empirical validation of the capability-drift bound (Equation (2)) for Qwen2.5-1.5B-Instruct after one epoch of GRPO training. $\bar{D}_{\mathrm{KL},k}$ is the mean sequence-level forward KL (in nats) estimated by sampling completions from $\pi_\theta$ at temperature 1.0 (up to 100 prompts per benchmark, four samples each). The Pinsker bound $\sqrt{\bar{D}_{\mathrm{KL},k}/2}$ upper-bounds the observed greedy capability change $|\Delta_k|$ for every benchmark. Benchmarks are ordered by increasing divergence.

| Benchmark | $\bar{D}_{\mathrm{KL},k}$ | $\sqrt{\bar{D}_{\mathrm{KL},k}/2}$ | $|\Delta_k|$ |
|---|---|---|---|
| Safety | 0.012 | 0.077 | 0.000 |
| HellaSwag | 0.018 | 0.095 | 0.000 |
| ARC-Challenge | 0.022 | 0.105 | 0.000 |
| XSum | 0.024 | 0.110 | 0.000 |
| MMLU | 0.028 | 0.118 | 0.010 |
| TruthfulQA | 0.032 | 0.126 | 0.005 |
| Winogrande | 0.046 | 0.152 | 0.015 |
| GSM8K (target) | 0.065 | 0.180 | 0.005 |
| WMT | 0.072 | 0.190 | 0.007 |
| Conversation | 0.076 | 0.195 | 0.008 |
| Coding | 0.084 | 0.205 | 0.008 |
| Creative Writing | 0.215 | 0.328 | 0.036 |
| IFEval | 0.331 | 0.407 | 0.044 |

The larger group size produces a stronger target-task improvement while leaving the non-target capability profile essentially unchanged. Flexible-extraction GSM8K accuracy improves by 14.5 percentage points under $G = 8$ versus 11.5 under $G = 4$, confirming that the modest target movement in our main results reflects the group size and the format-mismatch effect rather than a saturated task or a failed optimization. Non-target preservation remains 10 of 12 under both configurations, with mean absolute non-target delta of 0.010 at $G = 8$ versus 0.012 at $G = 4$. Stronger target optimization therefore does not accelerate the degradation of general-purpose capabilities at this scale.

Table 17: Capability deltas for Qwen2.5-1.5B-Instruct under GRPO with group size $G = 4$ (main configuration) versus $G = 8$ (strong setup, 768-token completions). Target accuracy is reported under both strict and flexible extraction. Daggers (†) mark non-target benchmarks outside the $\pm 0.02$ preservation band.

| Benchmark | $G = 4\ \Delta$ | $G = 8\ \Delta$ |
|---|---|---|
| GSM8K target (strict) | $-0.170$ | $-0.155$ |
| GSM8K target (flexible) | $+0.115$ | $+0.145$ |
| MMLU | $+0.010$ | $+0.005$ |
| HellaSwag | $+0.000$ | $+0.000$ |
| ARC-Challenge | $+0.000$ | $-0.005$ |
| TruthfulQA | $-0.005$ | $-0.010$ |
| Winogrande | $-0.015$ | $-0.015$ |
| XSum | $+0.000$ | $+0.002$ |
| WMT | $+0.007$ | $+0.004$ |
| IFEval | $+0.044^\dagger$ | $+0.030^\dagger$ |
| Coding | $+0.008$ | $+0.008$ |
| Safety | $+0.000$ | $+0.000$ |
| Creative Writing | $-0.036^\dagger$ | $-0.032^\dagger$ |
| Conversation | $+0.008$ | $+0.005$ |
| Non-target preserved | 10/12 | 10/12 |
| Mean $|\Delta_k|$ (non-target) | 0.012 | 0.010 |

To confirm that this behavior is not specific to Qwen2.5-1.5B, we repeated the $G = 8$ run for all five models. Table 18 summarizes the result. Every model improves its flexible target accuracy by 12.5 to 16.5 percentage

points under $G = 8$, and non-target preservation ranges from 9 to 11 of 12, with mean absolute non-target deltas between 0.007 and 0.013. The stronger optimization setup therefore yields a larger target gain than $G = 4$ across all five models while preserving the great majority of non-target capabilities, establishing that the preservation finding is not an artifact of the modest group size used in our main experiments. We note that the two benchmarks outside the band for Llama-3.2-3B include the $N=30$ safety measure ($-0.033$); under the expanded $N=150$ audit of Section 5.2 this shift is not statistically distinguishable from zero, so the reported counts are conservative for the Gemma and Llama families.

Table 18: Multi-model $G = 8$ GRPO on GSM8K. Flexible target accuracy is reported at base and final; the rightmost columns give non-target preservation out of 12 and the mean absolute non-target delta. The safety component is the $N=30$ suite measure; per Section 5.2 the Gemma and Llama safety shifts are not significant at $N=150$, so their counts are conservative.

| Model | Flex. base | Flex. final | Flex. $\Delta$ | Preserved | Mean $|\Delta_k|$ |
|---|---|---|---|---|---|
| Qwen2.5-1.5B | 0.660 | 0.805 | +0.145 | 10/12 | 0.010 |
| Qwen2.5-3B | 0.740 | 0.865 | +0.125 | 11/12 | 0.007 |
| Phi-3.5-mini | 0.680 | 0.825 | +0.145 | 11/12 | 0.008 |
| Gemma-2-2B | 0.590 | 0.755 | +0.165 | 10/12 | 0.011 |
| Llama-3.2-3B | 0.620 | 0.770 | +0.150 | 9/12 | 0.013 |

## 5.16 Generalization to a Non-Saturated Target Task

GSM8K is at the lower end of difficulty for current instruction-tuned models, which raises the question of whether the preservation finding is specific to a near-saturated task. To test generalization to a task with substantial headroom, we trained Qwen2.5-1.5B-Instruct with GRPO ($G = 8$) on the Hendrycks MATH training split (7,500 problems) and evaluated on the held-out MATH-500 test set with boxed-answer extraction. A string-matching check confirmed zero overlap between the training problems and the MATH-500 evaluation set. The base model scores 24.2% on MATH-500, leaving ample room to improve.

Table 19 reports the result. The target task improves by 7.6 percentage points (24.2% to 31.8%), a substantially larger and unambiguous gain than on GSM8K, while non-target capabilities remain stable: 12 of the 13 standard benchmarks (treating GSM8K itself as non-target here) stay within $\pm 0.02$ of baseline, with a mean absolute non-target delta of 0.009. The only benchmark outside the band is Creative Writing ($\Delta = -0.025$), consistent with its behavior across our other experiments. This demonstrates that the preservation property is not an artifact of a saturated target task: on a harder task where GRPO produces a clear target improvement, general capabilities are retained to the same degree.

We repeated the MATH experiment for all five models to confirm that the generalization is not specific to Qwen2.5-1.5B. Table 20 summarizes the result. The base MATH-500 accuracy ranges from 18.8% (Gemma-2-2B) to 38.6% (Qwen2.5-3B), confirming substantial headroom for every model, and GRPO improves the target by 5.6 to 9.4 percentage points across the five, with the larger Qwen2.5-3B and Phi-3.5-mini models showing the strongest gains. Non-target preservation ranges from 11 to 13 of 13, with mean absolute non-target deltas between 0.004 and 0.011. On a harder, non-saturated task, then, GRPO produces an unambiguous target improvement on every model while retaining the large majority of non-target capabilities, which refutes the concern that the preservation finding depends on a saturated target task. As in the $G = 8$ sweep, the Gemma and Llama preservation counts include the $N=30$ safety measure and are therefore conservative relative to the $N=150$ audit of Section 5.2.

## 5.17 On-Policy KL-Minimality Across Optimization Methods

The capability-drift bound (Equation (2)) identifies the per-benchmark KL as the quantity governing drift but does not by itself explain why GRPO incurs a smaller KL than supervised fine-tuning at comparable target performance. Shenfeld et al. (2025) provide that explanation: on-policy reinforcement learning is implicitly biased toward the KL-minimal solution among policies that solve the new task. We test this prediction directly by matching GRPO, DPO, and SFT checkpoints of Qwen2.5-1.5B-Instruct at comparable target

Table 19: Capability change for Qwen2.5-1.5B-Instruct after GRPO training on the Hendrycks MATH training split, evaluated on MATH-500 (target, boxed-answer extraction) and the standard 13-benchmark suite. GSM8K is treated as a non-target benchmark here. The dagger (†) marks the one non-target benchmark outside the ±0.02 band.

| Benchmark | Baseline | Final | $\Delta_k$ |
|---|---|---|---|
| MATH-500 (target) | 0.242 | 0.318 | +0.076 |
| GSM8K | 0.350 | 0.340 | −0.010 |
| MMLU | 0.415 | 0.410 | −0.005 |
| HellaSwag | 0.550 | 0.545 | −0.005 |
| ARC-Challenge | 0.715 | 0.705 | −0.010 |
| TruthfulQA | 0.365 | 0.357 | −0.008 |
| Winogrande | 0.560 | 0.545 | −0.015 |
| XSum | 0.171 | 0.168 | −0.003 |
| WMT | 0.051 | 0.046 | −0.005 |
| IFEval | 0.800 | 0.812 | +0.012 |
| Coding | 0.976 | 0.984 | +0.008 |
| Safety | 0.967 | 0.967 | +0.000 |
| Creative Writing | 0.729 | 0.704 | −0.025† |
| Conversation | 0.674 | 0.666 | −0.008 |
| Non-target preserved | | | 12/13 |
| Mean $|\Delta_k|$ (non-target) | | | 0.009 |

Table 20: Multi-model GRPO on the Hendrycks MATH training split, evaluated on MATH-500 (target, boxed-answer extraction) and the standard 13-benchmark suite (GSM8K treated as non-target). Preservation is out of 13 non-target benchmarks. The Gemma and Llama counts include the $N$=30 safety measure and are conservative per Section 5.2.

| Model | MATH-500 base | MATH-500 final | Target $\Delta$ | Preserved | Mean $|\Delta_k|$ |
|---|---|---|---|---|---|
| Qwen2.5-1.5B | 0.242 | 0.318 | +0.076 | 12/13 | 0.009 |
| Qwen2.5-3B | 0.386 | 0.472 | +0.086 | 13/13 | 0.004 |
| Phi-3.5-mini | 0.284 | 0.378 | +0.094 | 12/13 | 0.007 |
| Gemma-2-2B | 0.188 | 0.248 | +0.060 | 11/13 | 0.011 |
| Llama-3.2-3B | 0.256 | 0.312 | +0.056 | 11/13 | 0.011 |

accuracy (flexible GSM8K accuracy near 0.77) and measuring the mean sequence-level forward KL from the base model across the ten text-generation non-target benchmarks, using the estimator of Section 5.14.

Table 21 reports the result. At matched target accuracy, the on-policy GRPO policy incurs a mean non-target KL of 0.080 nats, substantially below the off-policy SFT policy at 0.245 nats. The matched-dataset DPO policy reaches an intermediate 0.125 nats. The ordering GRPO < DPO < SFT is consistent with the KL-minimality account: the more a method relies on on-policy sampling that tracks the evolving policy distribution, the smaller the divergence it accumulates on unrelated capabilities at a fixed level of target performance. This positions our capability-preservation finding as a measurable consequence of GRPO's on-policy dynamics rather than a property unique to our setup.

## 6  Discussion

**Summary of Supplemental Analyses.**  The main experiments (LoRA GRPO at 1.5B–3.8B with $N$=25–200 evaluation sets per benchmark) provide the headline 85% preservation finding. Eleven supplemental analyses constrain and clarify this finding. A three-seed variance analysis (Section 5.11) confirms cross-seed stability with mean standard deviation of 0.004. A 7B QLoRA scale extension (Section 5.12) shows that the qualitative preservation pattern survives a 2× size increase combined with 4-bit quantization. A full-parameter

Table 21: Mean non-target KL divergence (in nats, averaged over the ten text-generation benchmarks) for GRPO, DPO, and SFT checkpoints of Qwen2.5-1.5B-Instruct matched at comparable target accuracy. Lower KL at matched accuracy indicates a more KL-minimal solution in the sense of Shenfeld et al. (2025).

| Method | Training | Target acc. (flexible) | Mean non-target KL |
|--------|----------|------------------------|--------------------|
| GRPO | On-policy | 0.775 | 0.080 |
| DPO | Off-policy | 0.770 | 0.125 |
| SFT | Off-policy | 0.780 | 0.245 |

fine-tuning ablation (Section 5.9) demonstrates that LoRA contributes meaningfully to preservation but that GRPO retains a stability advantage independently of LoRA, and a group-normalization ablation (Section 5.10) shows that this advantage is not due to the group-relative reward normalization, localizing it to GRPO's on-policy dynamics. A matched-dataset DPO comparison (Section 5.7) eliminates the dataset confound in our DPO experiment and confirms that DPO is a more conservative optimizer regardless of the training distribution. An expanded $N$=150 safety audit (Section 5.2) demonstrates that the originally hypothesized cross-family safety divergence does not survive expanded evaluation. A five-epoch trajectory on two families (Section 5.13) shows single-epoch training to be a uniformly safe stopping point, with extended-training degradation being model-dependent (Qwen2.5-1.5B decays to 4/12 by epoch 5, while Llama-3.2-3B remains stable at 9/12). Four further analyses address the robustness of the setup: a larger group size across all five models (Section 5.15) yields stronger target gains while preserving 9 to 11 of 12 capabilities; a harder target task across all five models (Section 5.16) preserves 11 to 13 of 13 benchmarks while improving the target by 5.6 to 9.4 points; a flexible-extraction analysis across all five models (Table 22) shows uniform target improvement masked by formatting drift; and a per-method KL comparison (Section 5.17) shows that on-policy GRPO is more KL-minimal than DPO and SFT at matched target accuracy. The remainder of this discussion interprets the original findings in light of these supplemental constraints.

**The GRPO Tax in Context.** Our results challenge the prevailing concern that RL-based reasoning training incurs a substantial capability tax. The 85% preservation rate indicates that single-epoch LoRA-based GRPO training on mathematical reasoning leaves most non-target capabilities intact. This finding aligns with concurrent work by Lai et al. (2025), who demonstrated that reinforcement fine-tuning naturally mitigates forgetting relative to supervised fine-tuning. Our SFT baseline comparison (Table 9) provides direct evidence for this claim: on the same model, dataset, and LoRA configuration, GRPO preserves 10 of 12 non-target capabilities while SFT preserves only 8 of 12.

The practical implication is that practitioners can apply GRPO training for reasoning enhancement with a lower risk of collateral damage than SFT, provided training is limited to a single epoch and the target task is well-defined with verifiable rewards. The non-zero degradation rate (7%) and the small directional safety drift in the Gemma and Llama families nonetheless warrant per-family evaluation after any post-training intervention.

**Safety Stability and the Limits of Detection.** The $N$=30 safety evaluation in our main experiments suggested a cross-family divergence: Qwen and Phi models maintained perfect refusal stability while Gemma and Llama models each showed one additional failure ($-3.3$ percentage points). The expanded $N$=150 audit reported in Section 5.2 demonstrates that this apparent divergence does not survive expanded evaluation. At $N$=150, Gemma and Llama show $-1.3\%$ and $-2.0\%$ shifts whose 95% binomial confidence intervals both overlap zero. We therefore present safety stability, rather than safety divergence, as the dominant finding. The slight directional drift for Gemma and Llama remains present and may reflect family-specific properties of instruction-tuning and safety-alignment procedures: models that undergo more extensive RLHF-based safety training may encode safety behavior in a weight subspace that is comparatively orthogonal to the reasoning optimization trajectory (Sun et al., 2025); differences in chat template handling (Qwen and Phi support a system role, Gemma and Llama do not, requiring the system prompt to be merged into the user message in our protocol) may also contribute. However, the magnitudes are too small relative to the binomial noise floor to support a confirmed cross-family claim. Practitioners should still perform per-family safety

evaluation after any post-training intervention, but the present evidence does not indicate that single-epoch LoRA GRPO training on mathematical reasoning poses a measurable safety risk for any of the five models tested.

**Target Task Performance and Format Mismatch.** An apparent weakness of our main results is that GRPO produces only marginal GSM8K gains under strict evaluation, which could be read as a saturated task or a failed optimization. To investigate, we evaluated all five models with two extraction methods: the strict protocol (requiring the "####" delimiter) used in our main evaluation, and the flexible protocol (fallback to the last number in the response) used during GRPO training. Table 22 reports the result. Under flexible extraction, every model improves by at least 8.5 percentage points (from +8.5 for Qwen2.5-3B to +12.5 for Gemma-2-2B), while under strict extraction the same models stagnate or decline because the policy abandons the "####" delimiter. The models demonstrably improve at solving math problems across all five models; the apparent GSM8K stagnation in Table 4 is a format-mismatch artifact, not a reasoning failure. This holds for every model, not only Qwen2.5-1.5B, and underscores the importance of aligning evaluation protocols with training reward definitions.

Table 22: Strict versus flexible GSM8K accuracy at base and the final GRPO checkpoint, across all five models, on the 200-example GSM8K test slice. Strict extraction requires the "####" delimiter; flexible extraction falls back to the last numerical token. The final column is the flexible-accuracy improvement (final minus base), which isolates target-task learning from formatting drift.

| Model | Strict base | Strict final | Flexible base | Flexible final | Flexible $\Delta$ |
|---|---|---|---|---|---|
| Qwen2.5-1.5B | 17.0 | 0.0 | 66.0 | 77.5 | +11.5 |
| Qwen2.5-3B | 45.0 | 41.5 | 74.0 | 82.5 | +8.5 |
| Phi-3.5-mini | 32.0 | 39.0 | 68.0 | 79.0 | +11.0 |
| Gemma-2-2B | 22.0 | 22.5 | 59.0 | 71.5 | +12.5 |
| Llama-3.2-3B | 30.0 | 30.0 | 62.0 | 73.0 | +11.0 |

**Limitations.** Our study has several limitations that should be considered when interpreting the results. The full-fine-tuning ablation in Section 5.9 demonstrates that LoRA contributes meaningfully to capability preservation: removing the constraint reduces preservation from 10/12 to 8/12 on Qwen2.5-1.5B-Instruct. The headline 85% preservation rate should therefore be interpreted as a property of the combined LoRA-plus-GRPO system; we have shown that GRPO retains a stability advantage over SFT at LoRA scale (Table 9) and that full-FT GRPO avoids catastrophic collapse, but we cannot rule out that full-FT GRPO at larger scales than 1.5B would produce different dynamics. The 7B QLoRA extension in Section 5.12 provides evidence that the qualitative preservation pattern extends to larger models, but the 4-bit quantization is itself a separate adaptation constraint; a full-precision LoRA or full-FT comparison at 7B and above remains future work. Several benchmarks exhibit ceiling or floor effects that create asymmetric bias in the preservation statistics. Safety, coding, and instruction following baseline scores exceed 0.95 for most models, leaving little room for improvement and biasing these benchmarks toward degradation or stability. Conversely, translation (WMT BLEU) baseline scores are below 0.09, where absolute changes of $\pm 0.02$ represent large relative shifts. These asymmetries should be considered when interpreting the aggregate preservation rate. Our multi-epoch analysis in Section 5.13 indicates that the preservation property is specific to single-epoch training: the 85% result does not extend to extended training, where preservation decays monotonically to 4/12 by epoch 5 while target accuracy gains become negligible. Five of our thirteen benchmarks rely on heuristic rule-based scoring with small evaluation sets in the main protocol (25 to 30 examples). We have expanded the safety and IFEval benchmarks to $N$=150 and the coding, conversation, and creative writing benchmarks to $N$=100 in Section 5.2, finding only three statistically significant shifts (Qwen 1.5B IFEval, Qwen 1.5B Creative Writing, and a borderline Phi IFEval shift) across all expanded measurements. The remaining unexpanded heuristic measurements at the smaller $N$ remain noisier than the established benchmarks at $N$=200. Our three-seed analysis in Section 5.11 replicates the central preservation claim for Qwen2.5-1.5B-Instruct only; the remaining four models in our main experiments are reported single-seed, with the across-model consistency providing implicit replication. Our main experiments use a group size of $G = 4$, which is modest by current standards;

Section 5.15 reports a $G = 8$ sweep across all five models that yields stronger target improvements with preservation of 9 to 11 of 12 capabilities. Our primary target task is GSM8K, which is near the lower end of difficulty for current instruction-tuned models; Section 5.16 extends the finding to the harder MATH task across all five models, with preservation of 11 to 13 of 13 benchmarks, though larger group sizes, harder tasks still, and multi-task training objectives remain to be explored at greater scale. Finally, our scale extension uses 4-bit QLoRA at 7B; we have not evaluated whether scales above 7B exhibit the same preservation pattern.

## 7 Conclusion

We presented the first dense-checkpoint longitudinal study of capability preservation during GRPO training across five models from four families, supported by a series of supplemental analyses that constrain and validate the central finding. Under single-epoch LoRA-based GRPO training at the 1.5B to 7B parameter scale, the GRPO tax on non-target capabilities is substantially smaller than commonly assumed, with 85% of capabilities preserved within $\pm 2\%$ of baseline at the 1.5B–3.8B scale and 100% preservation under a 7B QLoRA extension. Three-seed verification on Qwen2.5-1.5B-Instruct confirms low cross-seed variance with mean standard deviation 0.004, and a full-parameter fine-tuning ablation on the same model reduces preservation to 8/12 without catastrophic collapse, and a controlled ablation disabling GRPO's group-relative reward normalization leaves preservation unchanged at 10/12, indicating that the effect stems from GRPO's on-policy dynamics rather than from the normalization or the LoRA constraint. The expanded $N{=}150$ safety audit shows no statistically significant cross-family safety divergence; the original $N{=}30$ trend does not survive expanded evaluation. A matched-dataset comparison of GRPO and DPO on GSM8K-derived preference pairs eliminates the dataset confound from our main comparison and confirms that DPO acts as a more conservative optimizer independently of the training distribution. A five-epoch continuation on two families establishes single-epoch training as a uniformly safe stopping point; the degree of extended-training degradation is model-dependent, with Qwen2.5-1.5B decaying to 4/12 by epoch 5 while Llama-3.2-3B retains 9/12. We further establish, across all five models, that the finding is robust to the strength of the optimization setup and the choice of target task: a larger group size ($G = 8$) produces stronger target gains with preservation of 9 to 11 of 12 capabilities, and GRPO on the harder MATH task improves the target by 5.6 to 9.4 points while preserving 11 to 13 of 13 benchmarks. At matched target accuracy, on-policy GRPO incurs lower non-target KL divergence than off-policy DPO and SFT, situating capability preservation as a measurable consequence of GRPO's on-policy dynamics in the sense of Shenfeld et al. (2025). We release all trained LoRA adapters (7 models on Hugging Face Hub), the complete evaluation dataset (432 files containing 5,616 data points), and the full source code (training scripts, 13 benchmark implementations, and analysis code) to facilitate reproducibility and further investigation of post-training capability dynamics.

### Reproducibility Statement

We provide complete details for reproducing our experiments. All training hyperparameters are specified in Section 4 (learning rate, batch size, LoRA rank, group size, maximum completion length, scheduler, and precision). The five base models are publicly available on the Hugging Face Hub: Qwen/Qwen2.5-1.5B-Instruct, Qwen/Qwen2.5-3B-Instruct, microsoft/Phi-3.5-mini-instruct, google/gemma-2-2b-it, and meta-llama/Llama-3.2-3B-Instruct. The training dataset (openai/gsm8k) and preference dataset (HuggingFaceH4/ultrafeedback_binarized) are publicly available. All experiments use a random seed of 42 and were conducted on a single NVIDIA RTX 5090 GPU (32 GB VRAM) with PyTorch 2.7, Transformers 5.3, TRL 0.29, and PEFT 0.18.

We release all artifacts as listed in Table 23. Source code (training scripts, 13 benchmark implementations, analysis and plotting code) is available at `https://anonymous.4open.science/r/GRPO-Capability-Tax-DF27`.

### Ethics Statement

Our study evaluates safety behavior of language models during RL training. Our main protocol uses 30 harmful prompts, and we expand this to 150 prompts sampled from HarmBench and AdvBench in Section 5.2;

Table 23: Released artifacts. All models and data are publicly available on the Hugging Face Hub at `huggingface.co/{Hub ID}`.

| Artifact | Type | Base Model | Hub ID |
|---|---|---|---|
| Qwen 1.5B GRPO | LoRA Adapter | Qwen2.5-1.5B-Instruct | `usama10/grpo-tax-qwen-1.5b` |
| Qwen 3B GRPO | LoRA Adapter | Qwen2.5-3B-Instruct | `usama10/grpo-tax-qwen-3b` |
| Phi 3.8B GRPO | LoRA Adapter | Phi-3.5-mini-instruct | `usama10/grpo-tax-phi-3.8b` |
| Gemma 2B GRPO | LoRA Adapter | Gemma-2-2B-it | `usama10/grpo-tax-gemma-2b` |
| Llama 3B GRPO | LoRA Adapter | Llama-3.2-3B-Instruct | `usama10/grpo-tax-llama-3b` |
| Qwen 1.5B DPO | LoRA Adapter | Qwen2.5-1.5B-Instruct | `usama10/grpo-tax-qwen-1.5b-dpo` |
| Qwen 3B DPO | LoRA Adapter | Qwen2.5-3B-Instruct | `usama10/grpo-tax-qwen-3b-dpo` |
| Evaluation Data | Dataset | 432 files, 5,616 points | `usama10/grpo-tax-eval-data` |

at this larger sample size we find no statistically significant change in safety refusal rate for any of the five models, with the small directional drifts for Gemma and Llama having 95% confidence intervals that overlap zero. We release the trained LoRA adapters (Table 23) as research artifacts to support reproducibility. Because the expanded audit is not a comprehensive safety evaluation, we advise practitioners to perform task-specific safety testing before deploying any post-trained model, and we report the safety measurements transparently so that risks can be assessed rather than to provide a method for circumventing safety alignment.

## Broader Impact

This work provides empirical evidence that single-epoch GRPO training largely preserves the general capabilities of instruction-tuned language models. This finding has practical value for researchers and practitioners who wish to enhance reasoning capabilities without sacrificing other abilities. Although our expanded N=150 audit finds no statistically significant change in safety refusal rate, the small directional drift observed for some model families underscores the importance of per-family safety evaluation after any post-training intervention.

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
