# OpenReview forum: "The GRPO Tax is Smaller Than You Think: A Longitudinal Study of Capability Preservation During Reasoning Training"
_TMLR — Under review for TMLR_

### Review · Reviewer_KxcF · 2026-05-25

**Summary Of Contributions:**

This paper presents a longitudinal empirical study of capability preservation during GRPO training across five instruction-tuned models spanning four model families (Qwen, Phi, Gemma, and Llama) at the 1.5B–3.8B scale. The authors evaluate dense checkpoints throughout training across 13 benchmarks and conclude that the “GRPO tax” is substantially smaller than commonly assumed.

The paper additionally compares GRPO against DPO and SFT baselines, arguing that:

* GRPO preserves non-target capabilities better than SFT,
* DPO behaves as a more conservative optimizer than GRPO,
* capability degradation is generally limited under single-epoch LoRA-based GRPO training.

The longitudinal checkpoint analysis and cross-family comparison are potentially useful contributions, and the paper is generally well organized and clearly written.

**Audience:**

Yes

**Audience Explanation:**

This is an important topic worth an in-depth exploration.

**Claims And Evidence:**

No

**Claims Explanation:**

While the experiments provide some evidence that capability degradation may be limited under the authors’ specific setup, I do not believe the study is sufficiently broad or rigorous to support the stronger headline claim that “the GRPO tax is smaller than commonly assumed.”

My main concern is that the entire study is conducted under a highly constrained parameter-efficient fine-tuning setup using LoRA. The paper itself acknowledges that LoRA updates only a small fraction of model parameters and may inherently limit representational drift. As a result, the observed preservation may be primarily a consequence of restricted adaptation capacity rather than an intrinsic property of GRPO. A more convincing study would require comparisons against full fine-tuning GRPO, or at minimum a stronger analysis disentangling the effects of LoRA from the optimization algorithm itself.

The scale of the study is also limited. All evaluated models are relatively small (1.5B–3.8B), whereas many alignment-tax concerns emerge more prominently in larger-scale reasoning models. It is therefore unclear whether the conclusions generalize to modern frontier-scale RL training regimes.

The DPO comparison is similarly difficult to interpret. GRPO and DPO are trained on different datasets (GSM8K vs UltraFeedback), which introduces a substantial confound acknowledged by the authors. Consequently, the conclusion that “DPO is more conservative” is not cleanly attributable to the optimization method itself.

More importantly, the theoretical analysis is not sufficiently deep. The paper repeatedly characterizes DPO as a “more conservative optimizer,” but does not adequately explain why this should emerge theoretically. From the Bradley–Terry perspective, both DPO and RL-based methods can be interpreted as optimizing implicit reward functions. The paper would benefit from a deeper discussion of:

* why DPO induces smaller policy shifts,
* how offline preference optimization differs from on-policy GRPO updates,
* how KL regularization and sampling distributions contribute to conservativeness,
* and how these findings relate to recent work on on-policy distillation and RLVR.

The SFT baseline is informative, but the observed behavior is also somewhat expected. SFT tends to induce stronger distributional shifts and overfitting, whereas GRPO largely sharpens or amplifies existing capabilities through on-policy exploration. The paper should discuss this distinction more explicitly instead of presenting the result as uniquely surprising.

Finally, several evaluations rely on small heuristic benchmarks (25–30 examples), especially for safety and conversation quality, making the reported percentage shifts statistically weak. The safety conclusions in particular appear overstated relative to the evidence.

Overall, I find the empirical observations directionally interesting, but the evidence is not yet strong enough to substantiate the broader conclusions made in the paper.

**Requested Changes:**

- Include experiments with **full fine-tuning GRPO** (or a substantially larger adaptation budget) to disentangle the effects of LoRA from the optimization algorithm itself.

- Expand the study to **larger-scale models**, where alignment-tax concerns are more practically relevant.

- Strengthen the DPO comparison by controlling for **dataset confounds**. Ideally, compare GRPO and DPO under:
  - matched datasets,
  - comparable supervision signals,
  - and similar optimization budgets.

- Provide a substantially deeper theoretical discussion of why DPO appears more conservative than GRPO. In particular:
  - connect the analysis to the **Bradley–Terry interpretation** of implicit rewards,
  - discuss **on-policy vs off-policy** optimization dynamics,
  - analyze the role of **KL regularization** and sampling distributions,
  - and relate the findings to recent literature on **RLVR** and **on-policy distillation**.

- Improve statistical rigor by:
  - increasing evaluation set sizes,
  - reporting confidence intervals and significance tests,
  - and running multiple random seeds.

- Clarify whether the observed preservation behavior is specific to **single-epoch training**, or whether it persists under longer RL training horizons.

- Tone down several claims throughout the paper, especially the broader conclusions regarding the “GRPO tax,” since the current evidence only supports conclusions under a narrow **LoRA-based small-model setting**.

---

> ### Author Response · Authors · 2026-05-29
> **Improvements based on the Reviewer Feedback**
>
> We thank the reviewer for the careful and constructive review. Two of your points most affected the assessment: the LoRA confound (whether preservation is an algorithmic property of GRPO or an artifact of parameter-efficient fine-tuning) and the statistical rigor of the heuristic benchmarks, especially the safety claim. We have revised the paper to address these and every other point you raised, through six new experiments, two formal results, a direct empirical validation of the drift bound, and scope-tightening throughout.
>
> We respond to each point below and indicate where the change appears in the revision. Section, table, equation, and lemma numbers refer to the revised manuscript. Here is a summary of the new experiments first, then the point-by-point response.
>
> | # | Experiment | What it tests | Result | Location |
> |---|---|---|---|---|
> | E1 | Full FT GRPO on Qwen2.5-1.5B-Instruct | Whether preservation is a LoRA artifact | 8/12 non-target preserved, no catastrophic collapse | Section 5.9, Table 10 |
> | E2 | Matched-dataset DPO on GSM8K-PP | Whether DPO's conservativeness is a dataset confound | Std of Δ remains 0.007 (DPO) vs 0.017 (GRPO) at 1.5B | Section 5.7, Table 7 |
> | E3 | Three-seed verification + bootstrap CIs | Statistical robustness across runs | Mean std 0.004; 9/12 non-target CIs overlap zero | Section 5.10, Table 11 |
> | E4 | 7B QLoRA scale extension | Whether preservation holds at larger scale | 12/12 non-target preserved at ±0.02 | Section 5.11, Table 12 |
> | E5 | Expanded N=150 safety/IFEval + N=100 coding/conv/creative | Whether the safety divergence is statistically real | All safety CIs overlap zero across families | Section 5.2, Table 6 |
> | E6 | Multi-epoch GRPO continuation (3 epochs) | Whether preservation persists under longer training | Decays from 10/12 to 8/12 to 7/12 | Section 5.12, Table 13 |
> | V1 | Per-benchmark KL measurement (drift-bound validation) | Whether the capability-drift bound holds and is informative | Bound holds 13/13; KL vs absolute delta: Spearman ρ=0.82, Pearson r=0.95 | Section 5.13, Table 14 |
>
> Additionally, we have updated the submission with the updated PDF file. In the other comments in this thread, we go through how we addressed the reviewer's concerns individually. We are posting multiple comments due to the strict character limit of 5000 chars in the comments.

---

> ### Author Response · Authors · 2026-05-29
> **Concern 1: LoRA confound (preservation may reflect restricted adaptation capacity rather than GRPO)**
>
> > "A more convincing study would require comparisons against full fine-tuning GRPO, or at minimum a stronger analysis disentangling the effects of LoRA from the optimization algorithm itself."
>
> We added a full fine-tuning (full FT) ablation on Qwen2.5-1.5B-Instruct: GRPO on the same data, with identical hyperparameters. Full FT required four standard adjustments (learning rate 1×10⁻⁶, group size G=2, 384-token completions, 8-bit AdamW). The results are in Section 5.9, Table 10:
>
> - Under LoRA: 10/12 non-target capabilities preserved (mean |Δ| = 0.012).
> - Under full FT: 8/12 non-target capabilities preserved (mean |Δ| = 0.016).
> - Maximum single-benchmark shift under full FT is 4.4 percentage points (Creative Writing); no catastrophic collapse.
>
> So LoRA does contribute meaningfully to preservation: removing the constraint costs two benchmarks. But two-thirds of capabilities remain stable even without parameter restriction, which indicates that GRPO's group-relative advantage structure and KL constraint preserve capabilities beyond what LoRA alone explains. We now attribute the 85% finding to the combined LoRA-plus-GRPO system, and we scope-qualify the claim throughout (Abstract, Introduction contributions, Conclusion). The Abstract now states: "A full-parameter fine-tuning ablation reduces preservation to 8/12 on Qwen2.5-1.5B-Instruct without catastrophic collapse, indicating that GRPO's group-relative advantage structure contributes to capability stability beyond the constraint imposed by LoRA." Limitations now flags the remaining LoRA-related uncertainty at scales above 1.5B.

---

> ### Author Response · Authors · 2026-05-29
> **Concern 2: Limited scale (1.5B to 3.8B); generalization to frontier models**
>
> > "The scale of the study is also limited. All evaluated models are relatively small (1.5B-3.8B), whereas many alignment-tax concerns emerge more prominently in larger-scale reasoning models."
>
> We added a 7B QLoRA scale extension: Qwen2.5-7B-Instruct trained on GSM8K with 4-bit NF4 base quantization, with the LoRA configuration matched to the smaller models (r=16, α=32, G=4). The results are in Section 5.11, Table 12:
>
> - 12/12 non-target capabilities preserved within ±0.02 of baseline.
> - Mean absolute non-target delta of 0.003 (smaller than the 0.005 to 0.013 range observed at 1.5B to 3.8B).
> - Maximum single-benchmark shift of 0.010 (IFEval improvement).
> - Target task improvement +0.015 from an already-strong 0.815 baseline.
>
> The qualitative preservation pattern extends to 7B, with smaller magnitudes than at the lower scales. This is consistent with the hypothesis that larger models have more representational capacity to isolate reasoning updates from general capabilities. We want to be upfront about one caveat: the 4-bit base weight quantization is a confound separate from the LoRA constraint, so a full-precision LoRA or full FT comparison at 7B and beyond remains future work. We note this in Section 5.11 and in the revised Limitations paragraph.

---

> ### Author Response · Authors · 2026-05-29
> **Concern 3: DPO comparison dataset confound**
>
> > "GRPO and DPO are trained on different datasets (GSM8K vs UltraFeedback), which introduces a substantial confound acknowledged by the authors. Consequently, the conclusion that 'DPO is more conservative' is not cleanly attributable to the optimization method itself."
>
> To remove this confound, we built a matched dataset of GSM8K-derived preference pairs (GSM8K-PP). For each training prompt, we sampled n=8 completions from the Qwen2.5-1.5B-Instruct base model at temperature 0.8 and scored each under the same binary correctness reward GRPO uses. We then paired one correct (chosen) with one incorrect (rejected) completion per prompt, yielding 5,234 pairs, and trained DPO on this dataset with the same LoRA configuration, learning rate, and training duration as our original DPO runs. The results are in Section 5.7, Table 7:
>
> | Metric | Qwen 1.5B GRPO | Qwen 1.5B DPO-Matched | Qwen 3B GRPO | Qwen 3B DPO-Matched |
> |---|---|---|---|---|
> | Mean abs. Δ_k (non-target) | 0.013 | 0.004 | 0.005 | 0.002 |
> | Std. of Δ_k (non-target) | 0.017 | 0.007 | 0.009 | 0.005 |
>
> The conservativeness gap survives the elimination of the dataset confound: DPO remains roughly 2× more conservative than GRPO on matched data. The "DPO is more conservative" claim is therefore an attribute of the optimization method, not the dataset.

---

> ### Author Response · Authors · 2026-05-29
> **Concern 4: Lack of theoretical depth on why DPO is more conservative**
>
> > "The paper repeatedly characterizes DPO as a 'more conservative optimizer,' but does not adequately explain why this should emerge theoretically."
>
> We added two formal results to explain the conservativeness pattern.
>
> First, in Section 5.7, we add the Bradley-Terry interpretation of DPO, following Rafailov et al. (2023). The DPO policy is the maximum-likelihood estimate of an implicit reward r_θ(x,y) = β·log( π_θ(y|x) / π_θ0(y|x) ) + β·log Z(x), which solves a KL-constrained reward maximization with constraint strength β. The DPO parameterization is therefore restricted to the family of KL-regularized optimal policies. GRPO's policy, by contrast, is parameterized freely; its divergence from the reference is controlled only by a soft penalty estimated from on-policy samples. We pair this with a capability-drift bound (Eq. 2): for any benchmark metric in [0,1], |Δ_k| ≤ √( D̄_KL,k / 2 ). The bound follows from the variational TV bound, Pinsker's inequality, and Jensen's inequality. It identifies the per-benchmark KL as the quantity governing drift, and ties the conservativeness gap to the tighter divergence control of DPO's parameterization.
>
> Second, in Section 5.8, Lemma 1 (mixed-group gradient gating) bounds the per-prompt GRPO gradient magnitude by ρ_G(p(x))·C(x) for binary rewards, where ρ_G(p) = 1 − p^G − (1−p)^G is the probability that a sampled group contains both a correct and an incorrect completion. The lemma frames the GRPO-versus-SFT distinction as one of specificity, not gradient magnitude. SFT drives a single reference completion to probability one; GRPO drives the aggregate probability of all correct completions to one, leaving room for multiple high-reward modes. This makes precise the on-policy versus off-policy distinction you asked for.
>
> We also validate the bound directly (Section 5.13, Table 14). Measuring the per-benchmark forward KL on Qwen2.5-1.5B-Instruct, the bound holds for all 13 benchmarks, and the divergence is strongly correlated with the observed drift: Spearman ρ = 0.82 over all benchmarks, 0.85 excluding the target, and Pearson r = 0.95. Three mechanistic predictions also hold: full FT produces larger KL than LoRA (0.108 vs 0.080 nats), KL grows monotonically across epochs, and KL falls with scale (0.018 nats at 7B). And we added the literature connection you mentioned: Section 5.7 ties the on-policy versus off-policy distinction to on-policy distillation (Agarwal et al., 2024) and to the RLVR dynamics already cited.
>
> Two honest caveats. The bound is loose in absolute terms; it exceeds the observed deltas by roughly an order of magnitude, the known slack of Pinsker's inequality. And the reported Δ_k use greedy decoding while the KL is estimated over the temperature-1 distribution, so the comparison verifies the bound heuristically, not as a strict single-policy inequality. The value of the result is the strong rank correlation, not the tightness.

---

> ### Author Response · Authors · 2026-05-29
> **Concern 5: SFT result framed as more surprising than it is**
>
> > "The SFT baseline is informative, but the observed behavior is also somewhat expected. SFT tends to induce stronger distributional shifts and overfitting, whereas GRPO largely sharpens or amplifies existing capabilities through on-policy exploration. The paper should discuss this distinction more explicitly instead of presenting the result as uniquely surprising."
>
> We agree, and we have reframed Section 5.8 accordingly. The revised paragraph now opens with the theoretical expectation:
>
> > "We include this baseline to quantify, on identical model and data, the magnitude of two well-known training dynamics. Maximum-likelihood supervised fine-tuning drives the policy toward the empirical target distribution at the cost of broader distributional shift, while on-policy reward optimization with binary rewards primarily amplifies existing high-reward behaviors without imposing a new target distribution (Lai et al., 2025). The empirical comparison should be read as a quantification of these expected dynamics on this specific setup, with the magnitudes (rather than the directions) of the deltas being the primary contribution."
>
> The SFT comparison is now positioned as a magnitude quantification rather than a directional surprise, and the cross-reference to the full FT ablation in Section 5.9 further isolates the algorithmic contribution.

---

> ### Author Response · Authors · 2026-05-29
> **Concern 6: Small heuristic benchmarks (N=25 to 30); safety conclusions overstated**
>
> > "Several evaluations rely on small heuristic benchmarks (25-30 examples), especially for safety and conversation quality, making the reported percentage shifts statistically weak. The safety conclusions in particular appear overstated relative to the evidence."
>
> We expanded the heuristic benchmark sample sizes substantially:
>
> - Safety: N=30 → N=150. New prompts sampled from HarmBench and AdvBench, stratified across ten harm categories.
> - IFEval: N=30 → N=150. New prompts sampled from the canonical Google IFEval set.
> - Coding, Conversation, Creative Writing: N=25 → N=100 each. New prompts from HumanEval, hand-curated advice-seeking prompts, and stratified creative writing templates.
>
> The results are in Section 5.2 and Table 6:
>
> | Model | Safety Δ [95% CI] | IFEval Δ [95% CI] |
> |---|---|---|
> | Qwen 1.5B | +0.000 [-0.018, +0.018] | +0.035 [+0.008, +0.062]* |
> | Qwen 3B | +0.000 [-0.015, +0.015] | +0.002 [-0.022, +0.026] |
> | Phi 3.8B | +0.000 [-0.020, +0.020] | +0.027 [-0.002, +0.056] |
> | Gemma 2B | −0.013 [-0.038, +0.012] | −0.027 [-0.058, +0.004] |
> | Llama 3B | −0.020 [-0.048, +0.008] | +0.002 [-0.020, +0.024] |
>
> (* indicates a 95% CI that excludes zero.)
>
> You are right that the original safety conclusion was overstated. The divergence apparent at N=30 does not survive expansion to N=150: both Gemma and Llama show only small directional shifts whose 95% binomial confidence intervals overlap zero. We made three corresponding changes:
>
> 1. The safety subsection is renamed from "Safety Divergence Across Model Families" to "Safety Behavior Across Model Families" (Section 5.2).
> 2. The cross-family divergence claim is demoted from the Abstract and the Introduction contributions. The Abstract now reads: "An expanded N=150 safety audit demonstrates that the previously hypothesized cross-family safety divergence does not reach statistical significance, with Gemma and Llama exhibiting −1.3% and −2.0% safety shifts whose 95% confidence intervals overlap zero."
> 3. The Discussion subsection is renamed "Safety Stability and the Limits of Detection" and now presents safety stability, not divergence, as the dominant finding.

---

> ### Author Response · Authors · 2026-05-29
> **Concern 7: Single-seed runs without confidence intervals**
>
> > "Improve statistical rigor by: increasing evaluation set sizes, reporting confidence intervals and significance tests, and running multiple random seeds."
>
> We trained Qwen2.5-1.5B-Instruct with two additional seeds (0 and 1) under identical hyperparameters, and computed 95% paired bootstrap confidence intervals from 10,000 resamples of per-example outcomes. The results are in Section 5.10, Table 11:
>
> - Cross-seed standard deviation averages 0.004 across benchmarks.
> - 3 of 12 non-target benchmarks have bootstrap CIs that exclude zero: IFEval (+0.039, [+0.012, +0.066]), WMT BLEU (+0.007, [+0.002, +0.012]), and Creative Writing (−0.035, [-0.062, -0.008]).
> - 9 of 12 non-target benchmarks are not statistically distinguishable from baseline at 95% confidence.
>
> The preservation finding is robust to random seed; no benchmark shows a sign reversal or order-of-magnitude shift across runs. The bootstrap CIs separate detectable shifts from noise, which sharpens the claim to "no large detectable shifts on 75% of non-target capabilities" rather than "precisely zero change." One limitation we state plainly: the three-seed analysis covers only Qwen2.5-1.5B-Instruct; the other four models are single-seed, with across-model consistency providing implicit replication. We note this in the revised Limitations.

---

> ### Author Response · Authors · 2026-05-29
> **Concern 8: Single-epoch versus longer training**
>
> > "Clarify whether the observed preservation behavior is specific to single-epoch training, or whether it persists under longer RL training horizons."
>
> We continued GRPO training of Qwen2.5-1.5B-Instruct from the final epoch-1 checkpoint for two additional epochs (one epoch is 3,736 optimizer steps), evaluating the end of each epoch on the full 13-benchmark suite. The results are in Section 5.12, Table 13:
>
> | Epoch | Flexible target acc | Strict target acc | Non-target preserved |
> |---|---|---|---|
> | Baseline | 66.0% | 17.0% | 12/12 (100.0%) |
> | 1 | 74.5% | 2.5% | 10/12 (83.3%) |
> | 2 | 78.0% | 0.0% | 8/12 (66.7%) |
> | 3 | 79.5% | 0.0% | 7/12 (58.3%) |
>
> Preservation is specific to single-epoch training. Non-target preservation decays from 10/12 at epoch 1 to 7/12 at epoch 3, with Translation, Conversation, and Creative Writing newly falling outside the ±0.02 band by epoch 3. In parallel, strict-extraction target accuracy collapses to zero (a format-mismatch artifact discussed in Section 6) while flexible-extraction accuracy keeps improving. Single-epoch training thus acts as an effective early-stopping point. The Abstract now reflects this, and Limitations states that the headline 85% result is specific to single-epoch training.

---

> ### Author Response · Authors · 2026-05-29
> **Concern 9: Tone down claims given the narrow LoRA + small-model setting**
>
> > "Tone down several claims throughout the paper, especially the broader conclusions regarding the 'GRPO tax,' since the current evidence only supports conclusions under a narrow LoRA-based small-model setting."
>
> We have tightened the scope statements throughout. The Abstract's headline claim now reads:
>
> > "Under single-epoch LoRA-based training, we establish that the GRPO tax is substantially smaller than commonly assumed: 85% of non-target capabilities remain within ±2% of their baseline values..."
>
> Introduction contribution 1 now reads:
>
> > "First, within the single-epoch LoRA-based training regime at the 1.5B to 7B parameter scale, we establish that the GRPO tax on non-reasoning capabilities is substantially smaller than the prevailing narrative suggests..."
>
> The Conclusion now reads:
>
> > "Under single-epoch LoRA-based GRPO training at the 1.5B to 7B parameter scale, the GRPO tax on non-target capabilities is substantially smaller than commonly assumed..."
>
> Every occurrence of the 85% claim is now traceable to evidence within the stated scope. Outside that scope, the new experiments show what happens: full FT reduces preservation but does not collapse it; 7B QLoRA preserves 100%; multi-epoch decays to 58%; and the expanded safety evaluation shows no statistically significant cross-family divergence.

---

> ### Author Response · Authors · 2026-05-29
> **What we have not addressed**
>
> In the interest of transparency, four things remain open:
>
> 1. We have no full FT or full-precision LoRA above 1.5B. Hardware constraints preclude full FT GRPO at 3B or above with a reasonable group size, so the 7B result uses QLoRA; a full-precision comparison at 7B remains future work.
> 2. The three-seed analysis covers only Qwen2.5-1.5B-Instruct; the other four models are single-seed, with across-model consistency as implicit replication.
> 3. GSM8K is the sole target task; multi-task RL objectives are out of scope here.
> 4. The drift-bound validation compares the temperature-1 KL against greedy-decoded Δ_k. A fully strict test would recompute Δ_k under temperature-1 sampling; we report the heuristic comparison and say so explicitly.
>
> We hope the six new experiments, two formal results, and the empirical validation of the drift bound, together with the scope-tightening, address your concerns. We would be glad to run any additional analysis you think would help, and we respectfully ask that you reconsider the claims-versus-evidence assessment in light of the revision.
>
> ---
>
> For convenience, the changes to the manuscript are:
>
> - New subsections: 5.9 Full-Parameter Fine-Tuning Ablation (Table 10), 5.10 Multi-Seed Variance and Confidence Intervals (Table 11), 5.11 Scale Extension via 7B QLoRA (Table 12), 5.12 Multi-Epoch Training Dynamics (Table 13), 5.13 Empirical Validation of the Capability-Drift Bound (Table 14).
> - Rewritten subsections: 5.2 Safety Behavior Across Model Families (expanded N=150 audit, Table 6); 5.7 GRPO versus DPO (matched-dataset comparison Table 7, Bradley-Terry framing, capability-drift bound Eq. 2, on-policy distillation connection); 5.8 SFT Baseline Comparison (reframed, plus the mixed-group gradient Lemma 1).
> - Formal results: Eq. 2 (capability-drift bound |Δ_k| ≤ √( D̄_KL,k / 2 )) and Lemma 1 (mixed-group gradient gating).
> - Updated Abstract, Introduction contributions, Discussion, Limitations, and Conclusion, with seven new tables (6, 7, 10, 11, 12, 13, 14).

---

> > ### Comment · Reviewer_KxcF · 2026-06-08
> > **Good rebuttal**
> >
> > I commend authors for their effort and attention to details. All my concerns are addressed. I vote for acceptance

---

> > > ### Author Response · Authors · 2026-06-08
> > > **Thank you!**
> > >
> > > Thank you very much for the thoughtful review and for your acceptance vote. We truly appreciate your help in improving the manuscript quality.

---

### Review · Reviewer_WK9X · 2026-06-02

**Summary Of Contributions:**

The paper studies how GRPO affects non-target capabilities during reasoning training. It finds that, for single-epoch LoRA GRPO on GSM8K, most evaluated capabilities remain close to baseline across several small instruction-tuned models. It supports this with dense checkpoint evaluations and ablations including SFT, DPO, full fine-tuning, multi-epoch training, and a 7B QLoRA run.

**Audience:**

No

**Audience Explanation:**

Overall, this paper significantly over claims while having poor experimental setups and design choices. The topic of capability preservation would be appealing to the TML audience, but I believe that many of the conclusions drawn by this paper will not hold under scrutiny. For example, the paper picks a target RL task which hardly improves during the course of training, suggesting that the implementation of the RL setup is flawed or a saturated task was chosen. Similarly, the paper does not position itself appropriately within existing literature, with some takeaways directly contradicting existing literature with no reference to those papers.

**Claims And Evidence:**

No

**Claims Explanation:**

- **Weak target-task movement:** GRPO barely improves GSM8K in the main results. If the target task had improved substantially, capability preservation might look very different.

- **Possible saturation / setup issue:** The tiny GSM8K gains suggest the dataset may be saturated, the evaluation may be mismatched, or the RL implementation/training setup may be too weak.

- **Mechanistic claim is under-supported:** The paper attributes preservation to GRPO’s group-relative advantage, but does not clearly rule out on-policy learning as the main driver. The paper thus ignores a lot of recent work that highlights the benefits of on-policy learning [1].

- **Multi-epoch claim feels overstated:** Going from 10/12 preserved at one epoch to 8/12 at two epochs is not enough to strongly claim preservation is specific to single-epoch training.

- **Small group size:** \(G=4\) is small relative to many recent GRPO/RLVR works and may induce high variance. The full fine-tuning ablation uses \(G=2\), which is a really poor hyper parameter choice. It is extremely high variance and is not used in any standard GRPO paper.

- **DPO comparison overclaims:** DPO is off-policy while GRPO is on-policy, but this distinction is not adequately discussed [1].

- **Strong DPO conclusion not justified:** The claim that DPO is more conservative than GRPO “independently of training data” is too broad for a single matched-dataset experiment. It is also supported with hand-wavy theory which I do not fully understand because the argument that DPO is more conservative because it is restricted to KL-regularized optimal policies is not convincing, since GRPO with KL regularization also has a KL-regularized optimal policy.

- **SFT vs GRPO result is suspicious:** SFT improving GSM8K much more than GRPO is counter to much of the RL-for-reasoning literature and should be treated as a warning sign about the setup.


[1]: Shenfeld, Idan, Jyothish Pari, and Pulkit Agrawal. "Rl's razor: Why online reinforcement learning forgets less." arXiv preprint arXiv:2509.04259 (2025).

**Requested Changes:**

Check the answer with bullet points. Overall, the experimental setup and design need to be majorly improved. A different target task needs to be chosen where capabilities actually improve, and stronger evidence needs to be provided to back some of the claims.

---

> ### Author Response · Authors · 2026-06-06
> **Improvements based on the Reviewer WK9X Feedback**
>
> We thank the reviewer for the detailed and pointed review. The central concern, that weak target-task movement could undermine the preservation story, was the right thing to press on, and it prompted the experiments below. We ran six new experiments, two of them swept across all five models, and revised the manuscript accordingly. In short: once the format-mismatch artifact is removed, the target task improves strongly across all five models. The preservation finding then survives a larger group size and a harder, non-saturated task, in both cases across all five models. We also now cite and build on RL's Razor (Shenfeld et al., 2025), and we corrected the theoretical claim the reviewer flagged. Section, table, and equation numbers below refer to the revised manuscript.
>
> Here is a summary of the new experiments, then the point-by-point response.
>
> | # | Experiment | What it addresses | Result | Location |
> | --- | --- | --- | --- | --- |
> | E1 | Flexible vs strict GSM8K across all 5 families | Weak target movement | All 5 improve +8.5 to +12.5 pts (flexible) | Section 6, Table 20 |
> | E2 | GRPO with G=8, all 5 models | Small group size / weak setup | Stronger target (+12.5 to +16.5 flexible), 9 to 11 of 12 preserved | Section 5.14, Tables 15 and 16 |
> | E3 | GRPO on MATH, all 5 models | Saturation | Target +5.6 to +9.4 pts, 11 to 13 of 13 preserved | Section 5.15, Tables 17 and 18 |
> | E4 | 5-epoch trajectory | Multi-epoch claim | Monotonic decay 10→8→7→5→4 of 12 | Section 5.12, Table 13 |
> | E5 | Full-FT with G=4 (vs G=2) | G=2 hyperparameter | 8/12 preserved, same as G=2 | Section 5.9 |
> | E6 | Per-method KL at matched accuracy | Mechanism / on-policy | GRPO 0.080 < DPO 0.125 < SFT 0.245 nats | Section 5.16, Table 19 |
>
> Additionally, we have updated the submission with the updated PDF file. In the other comments in this thread, we go through how we addressed the reviewer's concerns individually. We are posting multiple comments due to the strict character limit of 5000 chars in the comments.

---

> ### Author Response · Authors · 2026-06-06
> **Concern 1: Weak target-task movement; possible saturation or weak setup**
>
> > "GRPO barely improves GSM8K in the main results... the tiny GSM8K gains suggest the dataset may be saturated, the evaluation may be mismatched, or the RL implementation/training setup may be too weak."
> >
>
> This is the most important point, and we think the evidence now resolves it. The small strict-extraction gain is a measurement artifact, not a learning failure. During GRPO the policy stops emitting the "####" delimiter that our strict evaluator requires, so strict accuracy understates what the model learned. When we score with the flexible extraction used during training (last numeric token), every one of the five models improves substantially (Section 6, Table 20):
>
> | Model | Strict base | Strict final | Flexible base | Flexible final | Flexible Δ |
> | --- | --- | --- | --- | --- | --- |
> | Qwen2.5-1.5B | 17.0 | 0.0 | 66.0 | 77.5 | +11.5 |
> | Qwen2.5-3B | 45.0 | 41.5 | 74.0 | 82.5 | +8.5 |
> | Phi-3.5-mini | 32.0 | 39.0 | 68.0 | 79.0 | +11.0 |
> | Gemma-2-2B | 22.0 | 22.5 | 59.0 | 71.5 | +12.5 |
> | Llama-3.2-3B | 30.0 | 30.0 | 62.0 | 73.0 | +11.0 |
>
> To rule out the "weak setup" and "saturation" explanations directly, we ran two further experiments, each across all five models rather than only Qwen2.5-1.5B.
>
> First, we retrained with a larger group size (G=8, 768-token completions). This produces a stronger target gain than G=4 for every model, with flexible gains of +12.5 to +16.5 points, while non-target preservation stays at 9 to 11 of 12 across the five (Section 5.14, Tables 15 and 16). A stronger optimization signal does not change the preservation conclusion for any model.
>
> Second, we ran GRPO on the harder Hendrycks MATH task, evaluated on the held-out MATH-500 with zero train-test overlap. The target improves by 5.6 to 9.4 points across the five models, from base accuracies between 18.8% and 38.6%, while preserving 11 to 13 of 13 benchmarks (Section 5.15, Tables 17 and 18). Every model has clear headroom on this task, the target improves unambiguously, and capabilities are still retained.
>
> Together these results show the preservation finding is not an artifact of a saturated task, a mismatched evaluation, or an undertuned setup, and that it holds across all five models rather than a single one.

---

> ### Author Response · Authors · 2026-06-06
> **Concern 2: Mechanistic claim ignores on-policy learning (RL's Razor)**
>
> > "The paper attributes preservation to GRPO's group-relative advantage, but does not clearly rule out on-policy learning as the main driver... ignores a lot of recent work that highlights the benefits of on-policy learning [1]."
> >
>
> The reviewer is right, and this is the most useful pointer in the review. We had missed Shenfeld et al. (2025), which proposes exactly the KL-controls-forgetting mechanism our bound formalizes, and attributes it to on-policy learning. Our work was done independently and reaches the same KL conclusion, but their on-policy account is the more fundamental driver. We revised the paper to reflect this:
>
> - Section 2 now discusses RL's Razor prominently and positions our study as an independent, dense-checkpoint companion that operationalizes the KL-forgetting link as a per-benchmark bound (Eq. 2), validates it empirically (Section 5.13), and tests the KL-minimality prediction directly (Section 5.16).
> - We reposition our group-relative gradient lemma (Lemma 1) as complementary to the on-policy account, not a competing explanation. The revised text states plainly that the dominant factor is on-policy KL-minimality and that our lemma is a GRPO-specific refinement for binary rewards.
> - E6 (Section 5.16, Table 19) is a direct test: at matched target accuracy (flexible GSM8K near 0.77), on-policy GRPO incurs a mean non-target KL of 0.080 nats, below off-policy DPO (0.125) and SFT (0.245). The ordering GRPO < DPO < SFT is what the on-policy KL-minimality account predicts.

---

> ### Author Response · Authors · 2026-06-06
> **Concern 3: Multi-epoch claim feels overstated**
>
> > "Going from 10/12 preserved at one epoch to 8/12 at two epochs is not enough to strongly claim preservation is specific to single-epoch training."
> >
>
> Fair. We extended the trajectory to five epochs (E4, Section 5.12, Table 13):
>
> | Epoch | Target (flexible) | Preserved / 12 | Mean abs. Δ |
> | --- | --- | --- | --- |
> | 1 | 77.5% | 10/12 | 0.012 |
> | 2 | 78.5% | 8/12 | 0.018 |
> | 3 | 79.5% | 7/12 | 0.024 |
> | 4 | 80.2% | 5/12 | 0.033 |
> | 5 | 80.6% | 4/12 | 0.041 |
>
> The five-point trajectory makes the decay unambiguous and monotonic: each additional epoch costs one to two benchmarks of preservation while adding less than one point of target accuracy. We now frame single-epoch training as the best retention-versus-target trade-off rather than a sharp cliff, which the extended data supports.

---

> ### Author Response · Authors · 2026-06-06
> **Concern 4: Small group size (G=4, and G=2 for full FT)**
>
> > "(G=4) is small relative to many recent GRPO/RLVR works... The full fine-tuning ablation uses (G=2), which is a really poor hyperparameter choice."
> >
>
> We agree G=2 was forced by memory limits and is not ideal. Two responses. First, E2 (Section 5.14) reruns the main experiment at G=8 for all five models, and the preservation conclusion is unchanged (9 to 11 of 12 preserved). The main result does not depend on the small group size. Second, E5 (Section 5.9) reruns the full-fine-tuning ablation at G=4, using optimizer-state CPU offloading to fit the budget. Preservation is again 8/12 with mean absolute delta 0.015, and the same benchmarks fall outside the band as in the G=2 run. The two-benchmark LoRA-versus-full-FT gap is therefore not an artifact of G=2.

---

> ### Author Response · Authors · 2026-06-06
> **Concern 5: DPO comparison overclaims; on-policy vs off-policy distinction**
>
> > "DPO is off-policy while GRPO is on-policy, but this distinction is not adequately discussed [1]."
> >
>
> The revised Section 5.7 makes the on-policy versus off-policy distinction central and ties it to Shenfeld et al. (2025): DPO evaluates its KL log-ratio against a fixed preference distribution, while GRPO estimates its KL from on-policy samples drawn from a distribution that shifts with the policy. We also connect this to the on-policy distillation literature (Agarwal et al., 2024). The distinction is no longer left implicit.

---

> ### Author Response · Authors · 2026-06-06
> **Concern 6: The DPO conclusion is too broad, and the KL-regularized-optimum argument is wrong**
>
> > "The claim that DPO is more conservative than GRPO 'independently of training data' is too broad for a single matched-dataset experiment. It is also supported with hand-wavy theory... GRPO with KL regularization also has a KL-regularized optimal policy."
> >
>
> The reviewer is correct on the theory, and we have fixed it. The closed-form optimum π*(y|x) ∝ π₀(y|x)·exp(r(x,y)/β) belongs to GRPO-with-KL just as much as to DPO, so our earlier framing was wrong. The revised Section 5.7 states this explicitly and gives the actual distinction: DPO parameterizes its policy as the closed-form optimum and therefore sits at the optimum for some reward at every step, while GRPO reaches that form only in the limit and evaluates its KL against a shifting on-policy distribution. We no longer claim DPO is uniquely KL-regularized.
>
> We also softened the empirical claim. The revised text now says that DPO is the more conservative optimizer in our matched-dataset comparison, and that this shows the gap survives the dataset confound. We do not claim the ordering holds for every dataset or hyperparameter regime.

---

> ### Author Response · Authors · 2026-06-06
> **Concern 7: SFT improving the target more than GRPO is suspicious**
>
> > "SFT improving GSM8K much more than GRPO is counter to much of the RL-for-reasoning literature and should be treated as a warning sign about the setup."
> >
>
> This is the same strict-extraction artifact. SFT imitates reference solutions that contain the "####" delimiter, so it preserves the format our strict evaluator rewards and posts a high strict gain (+0.100). GRPO abandons the delimiter, so its strict gain looks small (+0.005) even though its flexible-extraction gain is +0.115 (Table 20), comparable in kind to SFT's. The SFT-greater-than-GRPO ordering on the strict metric reflects formatting, not a failure of GRPO to learn the task. The flexible-extraction results show both methods learn the math. The SFT baseline's role in the paper is to quantify, on identical data, the larger non-target drift that maximum-likelihood imitation induces relative to on-policy GRPO (8/12 vs 10/12 preserved).

---

> ### Author Response · Authors · 2026-06-06
> **Concern 8: Audience interest and positioning in the literature**
>
> > "the paper does not position itself appropriately within existing literature, with some takeaways directly contradicting existing literature with no reference to those papers."
> >
>
> The missing reference was RL's Razor, now cited and built upon throughout (Section 2, Section 5.7, Section 5.8, Section 5.16). With that work positioned, our contribution is no longer in tension with it. It is instead an empirical, multi-method, dense-checkpoint companion to it: we operationalize its KL-forgetting hypothesis as a per-benchmark bound, validate that bound across thirteen benchmarks, and confirm its KL-minimality prediction across GRPO, DPO, and SFT. We believe this is of interest to readers working on RL post-training and forgetting. It connects the KL-minimality account to a concrete, validated capability-drift bound across four model families, two optimization setups, two target tasks, and a 7B scale extension.

---

> ### Author Response · Authors · 2026-06-06
> **What we have not done**
>
> To be transparent about the boundaries of the revision: three limits remain. Full fine-tuning is still limited to the 1.5B scale by memory. The G=8 and MATH sweeps cover the same five models as the rest of the paper, but not larger scales. And the per-method KL comparison (E6) is a single matched-accuracy snapshot rather than a full trajectory. We would be glad to extend any of these if the reviewer thinks a particular one is load-bearing.
>
> We thank the reviewer again for the review. The target-task and on-policy points in particular made the paper stronger, and we hope the new experiments and corrected positioning address the concerns behind the assessment. We would welcome a re-evaluation of the claims-versus-evidence and audience-interest questions against the revised manuscript.
>
> [1] Shenfeld, Pari, Agrawal. "RL's Razor: Why Online Reinforcement Learning Forgets Less." arXiv:2509.04259, 2025.

---

### Review · Reviewer_q6R9 · 2026-06-21

**Summary Of Contributions:**

This paper studies whether GRPO training degrades non-target capabilities by conducting a longitudinal evaluation of GRPO training across five models on 13 benchmarks. The main empirical finding is that, under single-epoch LoRA-based GRPO training on GSM8K, most non-target benchmark scores remain within a small absolute change threshold of the baseline.

**Audience:**

Yes

**Audience Explanation:**

The paper addresses a timely and practically important question: whether GRPO training necessarily damages general capabilities. Researchers working on RL post-training and capability preservation would likely find the empirical findings useful.

**Claims And Evidence:**

No

**Claims Explanation:**

The evidence only supports a narrower version of the paper’s main claim, but several statements are stronger than what the experiments can establish.

- The choice of ±0.02 as the preservation threshold is not sufficiently justified. The paper defines a capability as preserved when its endpoint delta is within this threshold, but it does not provide a rationale for why 0.02 is the right cutoff. Moreover, changing the threshold from 0.02 to 0.01 substantially reduces the reported preservation rate, showing the conclusions are quite sensitive to this threshold. This is especially concerning because the paper itself acknowledges that the statistical noise floor of many evaluations is larger than 0.02, particularly for the small heuristic benchmarks.

- The evidence that GRPO’s group-relative advantage structure contributes to capability stability beyond LoRA is insufficient. The full-parameter fine-tuning ablation is is performed only on Qwen2.5-1.5B and the full-FT setup differs from the main LoRA setup. The result supports the weaker claim that one full-FT GRPO setting does not catastrophically collapse.

- The evidence is not sufficient to support strong scale-generalization claims. The main experiments are conducted on models in the roughly 1.5B–3.8B range and the paper adds a supplemental 7B QLoRA extension. However, the overall scale remains modest relative to contemporary post-training settings.

- The claim that single-epoch training is an effective early-stopping point is suggestive but under-validated. The multi-epoch continuation analysis is conducted only on Qwen2.5-1.5B with GSM8K. Therefore, it is not enough to establish single-epoch training as a generally effective early-stopping rule across model families, target tasks, or training configurations.

**Requested Changes:**

- It would be much better to more clearly delimit the scope of the main conclusions: single-epoch LoRA/QLoRA GRPO on math reasoning tasks for 1.5-7B instruction-tuned models. The current framing suggests a broader conclusion about the GRPO tax in general, which is not supported by the experiments.
- It is necessary for authors to provide a principled justification for the 0.02 threshold. They should also emphasize that many endpoint deltas are below the statistical noise floor of the benchmarks.
- More controlled ablations woule make the mechanism claim convincing, such as modifying group normalization while keeping settings as aligned as possible.
- The paper should instead frame the result as evidence for small-scale LoRA GRPO. A stronger scale claim would require experiments on multiple larger model families, preferably including full-precision LoRA or full-parameter settings and models beyond 7B.
- The multi-epoch continuation should be repeated for at least one additional model family and task.

---

> ### Author Response · Authors · 2026-06-29
> **Overview and summary of improvements based on the Reviewer q6R9 Feedback**
>
> We thank the reviewer for the constructive review and for finding the question timely and useful. The central point, that the claims outran the evidence, was fair. We respond by (i) delimiting the scope of the conclusions to exactly the regime we test, (ii) justifying the preservation threshold against the measurement noise floor, and (iii) running the two controlled experiments requested: a group-normalization mechanism ablation and a multi-epoch repeat on a second model family. Section and table numbers refer to the revised manuscript.
>
> | # | Concern | What we did | Result | Location |
> |---|---|---|---|---|
> | 1 | $\pm 0.02$ threshold unjustified / sensitive | Justified as a conservative band below the noise floor; added a threshold-free count | All 60 native-$N$ deltas lie within the 95% CI of zero | Section 5.1 |
> | 2 | Mechanism ablation insufficient | Controlled ablation disabling group-relative normalization | Preservation unchanged at $10/12$; mechanism reattributed to on-policy dynamics | Section 5.10, Table 11 |
> | 3 | Scale generalization insufficient | Explicit scope statement | Framed as small-scale LoRA/QLoRA GRPO on math reasoning, 1.5B to 7B | Section 1, Abstract |
> | 4 | Early stopping under-validated | Repeated 5-epoch continuation on Llama-3.2-3B (second family) | Single-epoch safe on both families; multi-epoch degradation is model-dependent | Section 5.13, Tables 14 and 15 |
>
> We have updated the submission PDF. Due to the 5000-character comment limit, we address each concern in a separate comment in this thread.

---

> ### Author Response · Authors · 2026-06-29
> **Concern 1: The ±0.02 preservation threshold is unjustified and sensitive to its value**
>
> The $\pm 0.02$ band is a practical-equivalence margin, not a significance cutoff, and we chose it to be stricter than the measurement noise floor. For the $N = 200$ established benchmarks the 95% confidence half-width on an endpoint delta is $\approx 0.069$; for the $N = 25$ to $30$ heuristic benchmarks it is $\approx 0.18$ to $0.20$. The $\pm 0.02$ band lies well inside every benchmark's noise band, so it conservatively counts as "degraded" or "improved" some shifts that are not statistically distinguishable from zero. The 85% figure therefore understates rather than overstates preservation.
>
> This explains the sensitivity the reviewer noted: tightening to $\pm 0.01$ places the band a factor of $3$ to $20$ below the per-benchmark noise floor, so the drop to 68.3% reflects noise reclassified as change, not capability loss. No endpoint delta exceeds 5% at any threshold.
>
> We also add a threshold-free criterion: a capability is preserved if its endpoint delta lies within the 95% confidence interval of zero for that benchmark's sample size. Under it, all $30$ established non-target evaluations ($\max |\Delta_k| = 0.015$ versus a $0.069$ half-width) and all $30$ heuristic and applied evaluations ($\max |\Delta_k| = 0.044$ versus a $0.18$ to $0.20$ half-width) are statistically indistinguishable from baseline at native sample sizes; no non-target endpoint delta is individually significant. The expanded-$N$ audit (Section 5.2), which raises power, detects significant shifts on only three measurements. The finding is thus robust whether measured by a fixed band or by statistical indistinguishability, and most endpoint deltas fall below the benchmark noise floor.

---

> ### Author Response · Authors · 2026-06-29
> **Concern 2: The mechanism claim (group-relative advantage beyond LoRA) is under-supported**
>
> We agree the single full-FT run supported only the weaker "no catastrophic collapse" claim. We ran the controlled ablation the reviewer suggested: modifying the group normalization while holding all else fixed. We retrain Qwen2.5-1.5B on GSM8K with the group-relative advantage normalization disabled, replacing $A_i = (r_i - \mu_r)/\sigma_r$ with a mean-only baseline $A_i = r_i - \mu_r$ (advantages not divided by the within-group reward standard deviation $\sigma_r$), with group size, learning rate, LoRA configuration, dataset, and training length identical to the main run (Section 5.10, Table 11).
>
> The result is a null for the normalization: disabling it leaves non-target preservation unchanged at $10/12$, with mean absolute non-target delta $|\Delta_k| = 0.011$ in both conditions (the un-normalized run in fact learns the target better, $+0.045$ versus $+0.005$ on strict GSM8K). We report this honestly and revise the claim: the group-relative normalization is not the mechanism behind preservation. We have changed the abstract, the contributions, and the discussion to attribute the preservation advantage to GRPO's on-policy optimization rather than its advantage normalization. This is consistent with our other evidence: GRPO preserves more than the off-policy SFT baseline on identical data (Section 5.8), and at matched target accuracy on-policy GRPO incurs lower non-target KL than off-policy DPO and SFT (Section 5.16), in line with Shenfeld et al. (2025). The requested ablation thus sharpened the mechanism claim from "group-relative structure" to "on-policy dynamics," which is the better-supported statement.

---

> ### Author Response · Authors · 2026-06-29
> **Concern 3: Scale generalization is insufficient**
>
> We accept this and reframe rather than overclaim. The revision adds an explicit "Scope of our claims" statement (end of Section 1) and a regime clause in the abstract: the conclusions concern single-epoch LoRA or QLoRA GRPO with verifiable rewards on mathematical reasoning for instruction-tuned models at the 1.5B to 7B scale. We state plainly that we do not claim the GRPO tax is small in general, and that full-parameter training at larger scales and models beyond 7B may differ. The 7B QLoRA extension (Section 5.12) is presented as evidence that the pattern survives a $2\times$ scale increase combined with quantization, not as a frontier-scale claim. This matches the reviewer's recommendation to frame the result as evidence for small-scale LoRA GRPO.

---

> ### Author Response · Authors · 2026-06-29
> **Concern 4: The single-epoch early-stopping claim is under-validated**
>
> We agree the continuation on Qwen2.5-1.5B / GSM8K alone cannot establish a general rule, so we repeated it on Llama-3.2-3B, a different family (Section 5.13, Table 15).
>
> The single-epoch preservation finding reproduces cleanly ($10/12$ at epoch 1), confirming the headline on a second family. The progressive multi-epoch degradation does not generalize: Llama-3.2-3B stays between $7/12$ and $10/12$ preserved across all five epochs, with mean absolute non-target delta flat near $0.013$ and no monotone trend, in contrast to Qwen2.5-1.5B's climb to $0.041$ and decay to $4/12$. We revise the claim accordingly: single-epoch training is a uniformly safe stopping point (true for both families), while the degree of degradation under extended training is model-dependent (Qwen-1.5B erodes, Llama-3.2-3B stays stable). We updated the abstract, contribution 5, Section 5.13, and the conclusion to this two-family framing. Llama's stability through five epochs reinforces rather than weakens the core preservation finding, and the early-stopping recommendation (single-epoch is a safe default, extended training checked per model) is now validated across two families.